# The mTORC1-mediated activation of ATF4 promotes protein and glutathione synthesis downstream of growth signals

Margaret E Torrence[1], Michael R MacArthur[1,2], Aaron M Hosios[1], Alexander J Valvezan[3], John M Asara[4], James R Mitchell[1,2], Brendan D Manning[1]*

[1]Department of Molecular Metabolism, Harvard T. H. Chan School of Public Health, Boston, United States; [2]Department of Health Sciences and Technology, Swiss Federal Institute of Technology (ETH) Zurich, Zurich, Switzerland; [3]Center for Advanced Biotechnology and Medicine, Department of Pharmacology, Rutgers Robert Wood Johnson Medical School, Piscataway, United States; [4]Division of Signal Transduction, Beth Israel Deaconess Medical Center and Department of Medicine, Harvard Medical School, Boston, United States

**Abstract** The mechanistic target of rapamycin complex 1 (mTORC1) stimulates a coordinated anabolic program in response to growth-promoting signals. Paradoxically, recent studies indicate that mTORC1 can activate the transcription factor ATF4 through mechanisms distinct from its canonical induction by the integrated stress response (ISR). However, its broader roles as a downstream target of mTORC1 are unknown. Therefore, we directly compared ATF4-dependent transcriptional changes induced upon insulin-stimulated mTORC1 signaling to those activated by the ISR. In multiple mouse embryo fibroblast and human cancer cell lines, the mTORC1-ATF4 pathway stimulated expression of only a subset of the ATF4 target genes induced by the ISR, including genes involved in amino acid uptake, synthesis, and tRNA charging. We demonstrate that ATF4 is a metabolic effector of mTORC1 involved in both its established role in promoting protein synthesis and in a previously unappreciated function for mTORC1 in stimulating cellular cystine uptake and glutathione synthesis.

*For correspondence:
bmanning@hsph.harvard.edu

## Introduction

Pro-growth signals in the form of growth factors, hormones, and nutrients impinge on cellular metabolic programs in a coordinated fashion involving both acute, post-translational regulation and transcriptional control of nutrient transporters and metabolic enzymes. The mechanistic target of rapamycin complex 1 (mTORC1) acts as a central point of integration for these signals and propagates a downstream metabolic response that increases anabolic processes while decreasing specific catabolic processes. Through a variety of downstream effectors, mTORC1 stimulates the synthesis of the major macromolecules comprising cellular biomass, including protein, lipid, and nucleic acids, along with metabolic and adaptive pathways that support this anabolic program (*Valvezan and Manning, 2019*).

A particularly interesting feature of this coordinated metabolic program downstream of mTORC1 is the co-opting of key nutrient-sensing transcription factors that are established to be activated, independent of mTORC1, in response to depletion of specific nutrients (*Torrence and Manning, 2018*). In their canonical roles, these transcription factors serve to mount an adaptive response by upregulating genes that allow cells to overcome the specific nutrient deficiency. Perhaps the best characterized of these transcription factors with dual regulation is the hypoxia-inducible factor 1 (HIF1) comprising the labile HIF1α protein heterodimerized with the aryl hydrocarbon receptor

**eLife digest** When building healthy tissue, the human body must carefully control the growth of new cells to prevent them from becoming cancerous. A core component of this regulation is the protein mTORC1, which responds to various growth-stimulating factors and nutrients, and activates the chemical reactions cells need to grow. Part of this process involves controlling 'nutrient-sensing transcription factors' – proteins that regulate the activity of specific genes based on the availability of different nutrients.

One of these nutrient-sensing transcription factors, ATF4, has recently been shown to be involved in some of the processes triggered by mTORC1. The role this factor plays in how cells respond to stress – such as when specific nutrients are depleted, protein folding is disrupted or toxins are present – is well-studied. But how it reacts to the activation of mTORC1 is less clear. To bridge this gap, Torrence et al. studied mouse embryonic cells and human prostate cancer cells grown in the laboratory, to see whether mTORC1 influenced the behavior of ATF4 differently than cellular stress.

Cells were treated either with insulin, which activates mTORC1, or an antibiotic that sparks the stress response. The cells were then analyzed using a molecular tool to see which genes were switched on by ATF4 following treatment. This revealed that less than 10% of the genes activated by ATF4 during cellular stress are also activated in response to mTORC1-driven growth.

Many of the genes activated in both scenarios were involved in synthesizing and preparing the building blocks that make up proteins. This was consistent with the discovery that ATF4 helps mTORC1 stimulate growth by promoting protein synthesis. Torrence et al. also found that mTORC1's regulation of ATF4 stimulated the synthesis of glutathione, the most abundant antioxidant in cells.

The central role mTORC1 plays in controlling cell growth means it is important to understand how it works and how it can lead to uncontrolled growth in human diseases. mTORC1 is activated in many overgrowth syndromes and the majority of human cancers. These new findings could provide insight into how tumors coordinate their drive for growth while adapting to cellular stress, and reveal new drug targets for cancer treatment.

nuclear translocator (ARNT or HIF1β). Oxygen depletion (i.e., hypoxia) results in the rapid stabilization of HIF1α and allows the HIF1 heterodimer to induce genes involved in glucose uptake, glycolysis, and angiogenesis to adapt to hypoxia and decrease mitochondrial respiration (*Nakazawa et al., 2016*). On the other hand, in response to upstream growth factor signaling pathways, activation of mTORC1 stimulates an increase in HIF1α protein synthesis, leading to elevated expression of HIF1 gene targets (*Brugarolas et al., 2003*; *Düvel et al., 2010*; *Hudson et al., 2002*; *Laughner et al., 2001*; *Thomas et al., 2006*; *Zhong et al., 2000*). The result is an mTORC1-mediated increase in glucose uptake and glycolysis even when oxygen is not limiting (e.g., normoxia), a process referred to as aerobic glycolysis, which can support the production of biosynthetic precursors in the form of glycolytic intermediates. Similarly, the sterol regulatory element (SRE)-binding protein (SREBP) family of transcription factors are independently regulated by both adaptive nutrient signals and growth signals controlling mTORC1. The SREBPs are canonically activated upon sterol depletion and induce expression of the enzymes required for de novo synthesis of fatty acid and sterol lipids (*Horton et al., 2002*). However, insulin and growth factor signaling can also induce lipid synthesis via mTORC1-stimulated activation of SREBP and its lipogenic gene targets (*Düvel et al., 2010*; *Owen et al., 2012*; *Peterson et al., 2011*; *Porstmann et al., 2008*). Recent studies have suggested that the regulation of nutrient-sensing transcription factors by mTORC1 signaling extends to the activating transcription factor 4 (ATF4) (*Adams, 2007*; *Ben-Sahra et al., 2016*; *Torrence and Manning, 2018*).

ATF4 is a basic leucine zipper (bZIP) transcription factor that is selectively translated in response to specific forms of cellular stress to induce the expression of genes involved in adaptation to stress (*Walter and Ron, 2011*). This adaptive program is referred to as the integrated stress response (ISR) and is initiated by stress-activated protein kinases, including general control nonderepressible 2 (GCN2) activated upon amino acid deprivation and protein kinase RNA-like endoplasmic reticulum kinase (PERK) activated by ER stress, among others, which phosphorylate eIF2α on Ser51

(*Harding et al., 2000*). Phosphorylation of eIF2α serves to globally attenuate mRNA translation to conserve amino acids and energy and decrease the cellular protein load as one adaptive measure to overcome these stresses (*Clemens, 1996*). Importantly, a small number of mRNAs, including that encoding ATF4, exhibit increased translation upon eIF2α-Ser51 phosphorylation (*Vattem and Wek, 2004*). The stress-induced increase in ATF4 leads to the expression of a canonical set of ATF4 target genes, including those involved in nonessential amino acid (NEAA) biosynthesis and amino acid transport, as part of the adaptive cellular response specific to stresses such as amino acid depletion (*Harding et al., 2003*).

ATF4 functions in heterodimers with other bZIP transcription factors and also co-regulates many of its target genes with additional transcription factors as part of the cellular stress response (*Newman and Keating, 2003*; *Wortel et al., 2017*). However, whether and how the many distinct upstream stresses that activate ATF4 influence its heterodimerization partners and the induction of specific sets of genes is not well understood.

While ATF4 is a major downstream effector of the ISR, evidence has emerged that ATF4 can also be activated by pro-growth signals that stimulate mTORC1 signaling (*Adams, 2007*; *Ben-Sahra et al., 2016*), and cis-regulatory elements for ATF4 binding are enriched in the promoters of mTORC1-induced genes (*Düvel et al., 2010*). Importantly, the mTORC1-mediated activation of ATF4 involves its increased translation in a manner that is independent of the ISR and phosphoryla-tion of eIF2α (*Ben-Sahra et al., 2016*; *Park et al., 2017*). These findings suggest that, similar to HIF1 and SREBP, ATF4 induction may be mobilized as part of the broader anabolic program downstream of mTORC1. Indeed, our previous findings indicate that mTORC1 promotes de novo purine synthe-sis, in part, through induction of mitochondrial one-carbon metabolism via ATF4 activation and expression of its gene target MTHFD2 (*Ben-Sahra et al., 2016*).

How the ATF4-dependent gene program compares between its adaptive role in the ISR and its activation as a downstream effector of mTORC1 signaling and whether ATF4 contributes to estab-lished or new functions of mTORC1 are unknown (*Figure 1A*). Here, we find that the mTORC1-ATF4 program represents a small subset of ATF4-dependent genes induced by ER stress and includes genes encoding the enzymes required for tRNA charging, NEAA synthesis, and amino acid uptake. Consistent with regulation of these enzymes by mTORC1 through ATF4, ATF4 contributes to the induction of protein synthesis downstream of mTORC1. We also find that mTORC1 signaling pro-motes glutathione synthesis through ATF4 and its specific regulation of the cystine transporter SLC7A11. Thus, ATF4 is an anabolic effector of mTORC1 signaling, necessary for both its canonical regulation of protein synthesis and its induced synthesis of glutathione, the most abundant antioxi-dant in cells.

## Results

### mTORC1 signaling activates a subset of ATF4-dependent genes also activated by the ISR

To identify the ATF4-dependent gene targets downstream of mTORC1, we compared the insulin-induced, rapamycin-sensitive transcripts between wild-type MEFs and those with biallelic loss of ATF4 via CRISPR/Cas9 gene deletion (see Materials and methods). Consistent with our previous studies (*Ben-Sahra et al., 2016*), insulin stimulated an increase in ATF4 protein in MEFs, which was decreased with rapamycin (*Figure 1B*, *Figure 1—figure supplement 1*). In parallel, these cells were treated with a time course of tunicamycin, an inhibitor of N-glycosylation that potently induces ER stress and an increase in ATF4, to identify ATF4 gene targets downstream of the ISR. RNA-seq anal-ysis revealed that 20% of transcripts (253 total) significantly upregulated upon insulin stimulation were significantly blocked in their induction with rapamycin treatment. Approximately 30% of these mTORC1-regulated genes (77 total) lost their insulin responsiveness with ATF4 deletion. In compari-son, 36% of transcripts significantly induced with tunicamycin treatment at 4 hr were dependent on ATF4 (774 total). Importantly, the expression of just 61 genes was found to overlap between these two modes of ATF4 regulation, being ATF4 dependent in response to both mTORC1 activation and the ISR (*Figure 1C*, *Figure 1—source data 1*).

The RNA-seq analysis demonstrated that only 8% of ATF4 gene targets induced by ER stress were also significantly stimulated by mTORC1 signaling (e.g., insulin induced and rapamycin

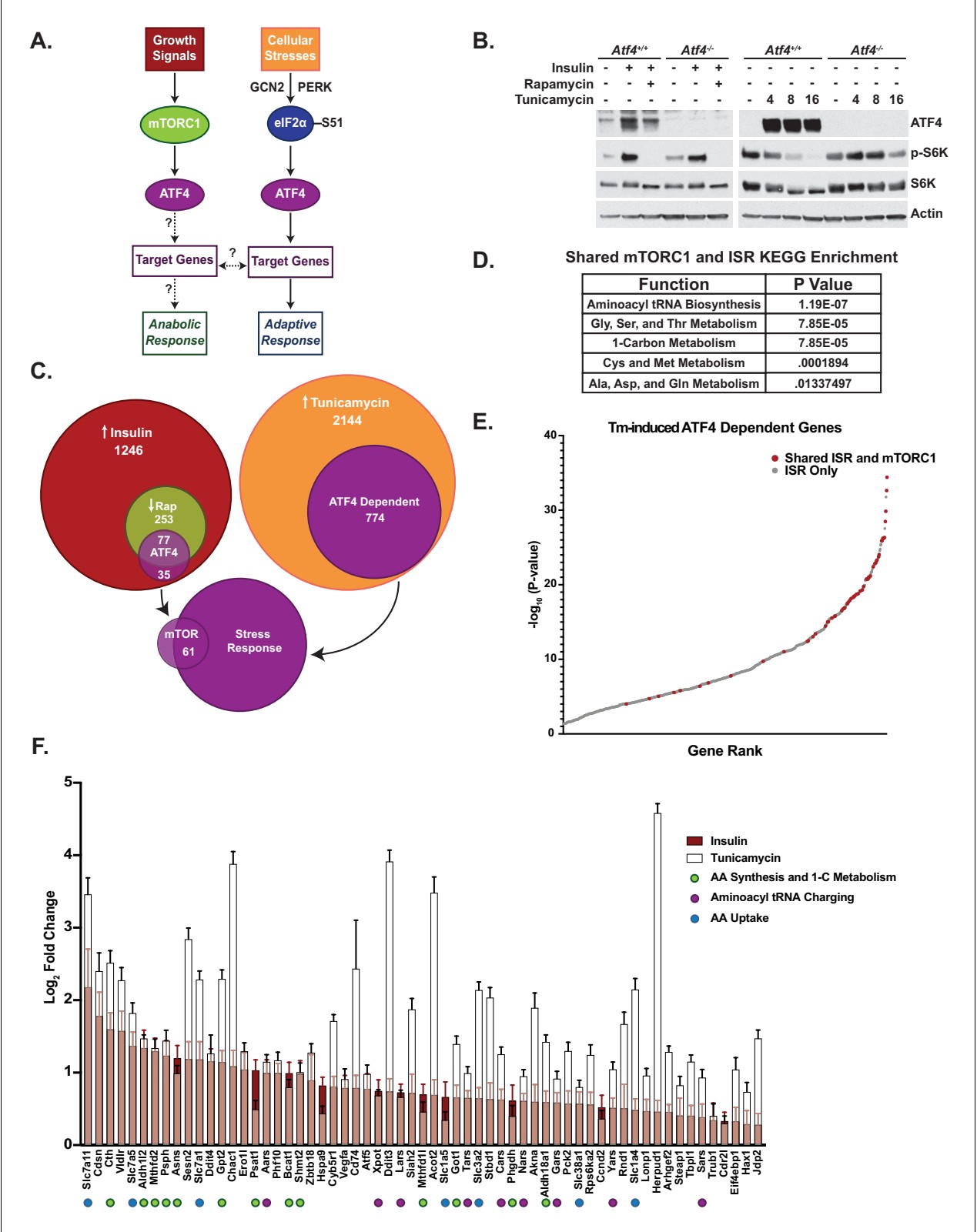

**Figure 1.** Mechanistic target of rapamycin complex 1 (mTORC1) signaling activates a subset of activating transcription factor 4 (ATF4)-dependent genes also activated by the integrated stress response (ISR). (**A**) Schematic of the dual regulation of ATF4 and the unknowns addressed in this study. (**B**) Immunoblots of parallel lysates from RNA-seq experiment. *Atf4*+/+ and *Atf4*-/- mouse embryo fibroblasts were treated, as indicated, with insulin (500 nM, 16 hr) or rapamycin (20 nM, 30 min) prior to insulin (left) or with tunicamycin (2 μg/mL) for 4, 8, or 16 hr (right). Insulin response is quantified in

*Figure 1 continued on next page*

*Figure 1 continued*

**Figure 1—figure supplement 1**. (C) Venn diagram depicting number and overlap of mTORC1- and ISR-induced transcripts, including those increased with insulin (red), decreased relative to insulin with rapamycin (green), and increased with 4 hr tunicamycin (orange), and those dependent on ATF4 within these categories (purple), all with p-values <0.05. Only 61 ATF4-dependent genes overlap between those significantly induced by insulin in a rapamycin-sensitive manner and those induced by tunicamycin. Gene lists per category are provided in *Figure 1—source data 1*. (D) KEGG enrichment of the shared mTORC1- and ISR-induced ATF4 target genes. p-Values provided were false discovery rate corrected. (E) Plot of -log$_{10}$p-values of 774 ATF4-dependent tunicamycin-induced genes. ATF4-dependent genes induced by both mTORC1 signaling and tunicamycin treatment (shared ISR and mTORC1) are shown in red. (F) The 61 ATF4-dependent genes induced by both mTORC1 (i.e., rapamycin-sensitive insulin stimulation) and tunicamycin treatment are shown ranked from left to right in order of greatest log$_2$-fold change with insulin (red bars), with the corresponding tunicamycin-induced changes superimposed (white bars) (n = 4). Error bars depict 95% confidence intervals.

The online version of this article includes the following source data and figure supplement(s) for figure 1:

**Source data 1.** Gene lists from *Figure 1C*.
**Figure supplement 1.** Quantification of immunoblot shown in *Figure 1B*.

sensitive). Interestingly, these 61 shared genes showed significant KEGG pathway enrichment for aminoacyl tRNA biosynthesis, amino acid metabolism, and one-carbon metabolism (*Figure 1D*). The genes shared between the ISR and mTORC1 signaling were greatly enriched among those exhibiting the most significant increase upon tunicamycin treatment, with 75% (46 genes) lying within the top 100 of the 774 tunicamycin-induced genes (*Figure 1E*). It is worth noting that many of the top ATF4-dependent genes that scored as being induced by the ISR alone showed some degree of rapamycin-sensitive induction with insulin but did not reach statistical significance in the RNA-seq analyses. These data indicate that the subset of ATF4-dependent genes induced by mTORC1 signaling largely comprised those that are also most sensitive to ATF4 induction by the ISR. Among the 61 shared ATF4-dependent transcripts, those involved in amino acid synthesis and transport, one-carbon metabolism, and aminoacyl tRNA charging often displayed comparable fold changes between insulin stimulation and tunicamycin treatment, while canonical genes of the ER stress response, such as *Herpud1* and *Ddit3/Chop,* showed much greater induction with tunicamycin (*Figure 1F*). In addition to aminoacyl tRNA synthetase genes, expression of the *Xpot* gene encoding Exportin-T, which is the major Ran GTPase family member for nuclear to cytosolic export of mature tRNAs (*Arts et al., 1998*; *Kutay et al., 1998*), was found to be similarly regulated by ATF4 in response to mTORC1 and ISR activation. Transcripts encoding known negative regulators of mTORC1 signaling are also among these 61 shared ATF4-induced genes, including *Ddit4/Redd1* and *Sesn2* (*Brugarolas et al., 2004*; *Condon et al., 2021*; *Lee et al., 2010*; *Reiling and Hafen, 2004*; *Wolfson et al., 2016*). These targets likely contribute to the ATF4-dependent inhibition of mTORC1 signaling (S6K1 phosphorylation) observed upon tunicamycin treatment (*Figure 1B*), while in the context of mTORC1 signaling, these ATF4 targets might play a role in negative feedback regulation of mTORC1. These data suggest that a specific subset of ATF4-dependent ISR-induced genes are likewise regulated by growth factor signaling through mTORC1 and are enriched for specific processes including aminoacyl tRNA synthesis, amino acid synthesis and uptake, and one-carbon metabolism.

ATF4 is known to form heterodimers with other bZIP transcription factors to engage its gene targets, while also co-regulating genes with transcription factors that bind additional promoter elements (*Kilberg et al., 2009*; *Newman and Keating, 2003*; *Wortel et al., 2017*). Thus, we used bioinformatic tools to determine whether the promoters of ATF4 gene targets shared between the ISR and mTORC1 signaling might be distinct from those induced by the ISR alone. Indeed, CiiiDER analysis (*Gearing et al., 2019*) revealed that there are predicted promoter-binding sequences that distinguish the 61 shared target genes from the top 200 ATF4-dependent genes induced by the ISR alone (*Figure 2A*, *Figure 2—source data 1*). Regulatory elements for the C/EBP family of transcription factors, which are well established to heterodimerize with ATF4 to induce its canonical downstream targets (*Cohen et al., 2015*; *Ebert et al., 2020*; *Huggins et al., 2015*), were the most enriched in the promoters of ATF4 gene targets with shared regulation. On the other hand, binding elements for the TEAD family of transcription factors, which function with YAP/TAZ in the Hippo signaling pathway, were enriched in the promoters of ATF4-dependent gene targets significantly induced only by tunicamycin, consistent with published work indicating a functional connection between the unfolded protein response and YAP-TEAD activation (*Wu et al., 2015*; *Xiao et al., 2019*). We next analyzed the 61 ATF4-dependent genes with shared regulation for physical evidence

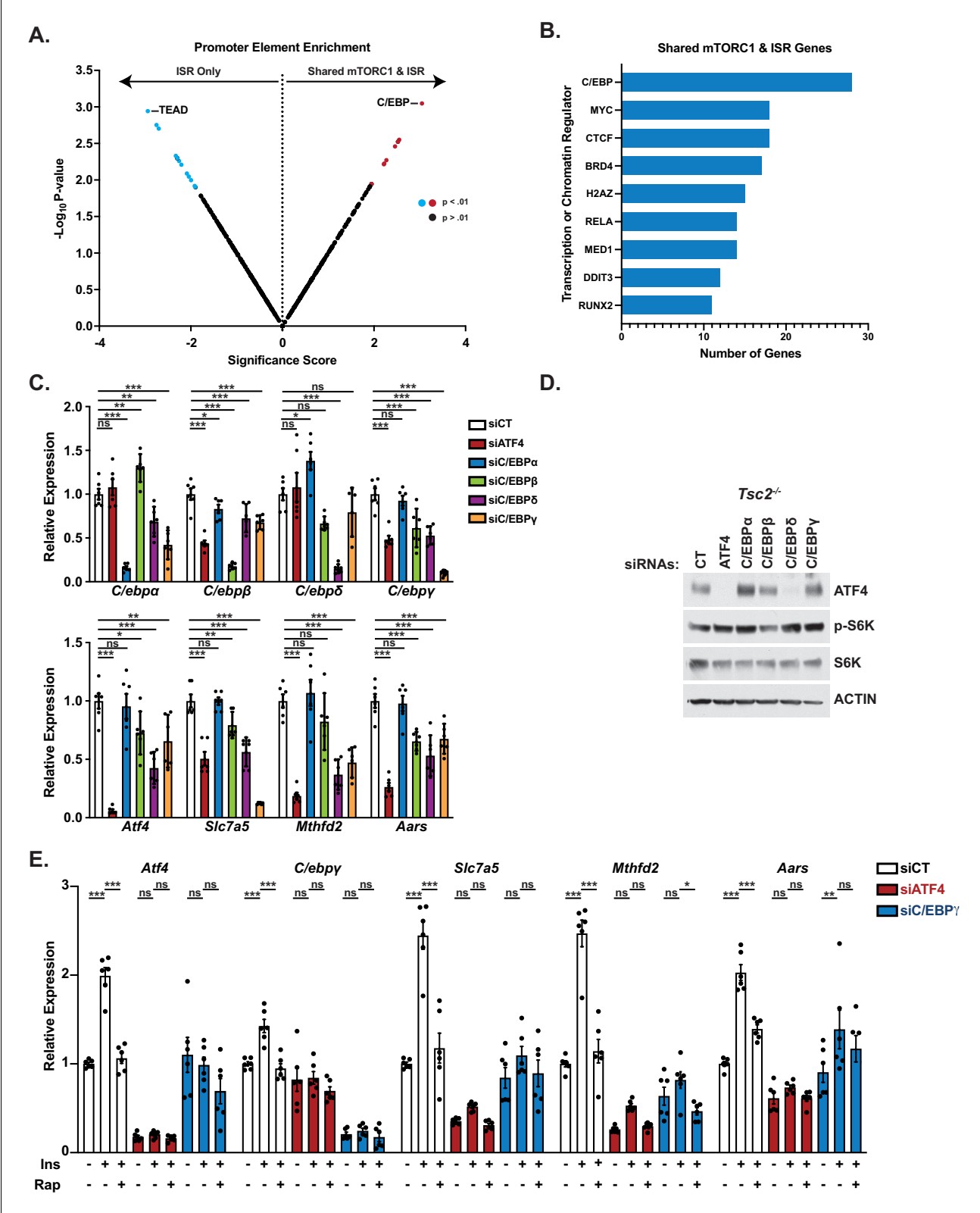

**Figure 2.** C/EBP family transcription factors contribute to the regulation of activating transcription factor 4 (ATF4)-dependent genes shared between mechanistic target of rapamycin complex 1 (mTORC1) signaling and the integrated stress response (ISR). (**A**) CiiiDER analysis comparing transcription factor-binding elements enriched in the promoters of the top 200 ATF4-dependent genes induced by tunicamycin but not insulin (ISR Only) versus the 61 ATF4-dependent genes induced by both mTORC1 signaling and tunicamycin (Shared ISR and mTORC1). Those sequence elements significantly
*Figure 2 continued on next page*

*Figure 2 continued*

enriched (p<0.01) are shown in blue or red. Data are provided in *Figure 2—source data 1*. (B) Cistrome analysis of genome-wide chromatin immunoprecipitation studies to identify transcription factors found to bind to the promoters of the ATF4-dependent genes shared in their regulation by mTORC1 and ISR. (C) qPCR analysis of the indicated transcripts in *Tsc2⁻/⁻* mouse embryo fibroblasts (MEFs) transfected with control siRNAs (siCT) or those targeting *Atf4, C/EBPα, C/EBPβ, C/EBPδ, C/EBPγ*. Expression relative to siCT for each gene is graphed as mean ± SEM from two independent experiments, each with three biological replicates (n = 6). (D) Immunoblots of cells treated as in (C). (E) qPCR analysis of the indicated transcripts in serum-deprived wild-type MEFs treated with insulin (500 nM, 16 hr) after 30 min pretreatment with vehicle or rapamycin (20 nM) following transfection with control siRNAs (siCT) or those targeting *Atf4* or *C/ebpγ*. Expression relative to siCT for each gene is graphed as mean ± SEM from two independent experiments, each with three biological replicates (n = 6). *p<0.05, **p<0.01, ***p<0.001, ns = not significant. One-way analysis of variance with Holm–Sidak method for multiple comparisons was used to determine statistical significance for (C, E).

The online version of this article includes the following source data for figure 2:

**Source data 1.** Promoter element enrichment data for *Figure 2A*.

of promoter binding of specific transcription factors using the Cistrome Data Browser, a portal for mining existing chromatin immunoprecipitation-DNA sequencing (ChIP-seq) data (*Mei et al., 2017*; *Zheng et al., 2019*). Importantly, this second unbiased analysis also revealed C/EBP isoforms as most commonly binding to the promoters of these genes (*Figure 2B*).

As all members of the C/EBP family have the potential to heterodimerize with ATF4 and contribute to the induction of these gene targets (*Newman and Keating, 2003*), we first determined the effects of siRNA-mediated knockdown of individual isoforms, relative to ATF4 knockdown, on expression of three representative genes in *Tsc2⁻/⁻* MEFs, which exhibit growth factor-independent activation of mTORC1 signaling. This analysis revealed that knockdown of ATF4, C/EBPβ, C/EBPδ, or C/EBPγ each led to decreased transcript levels of the shared mTORC1 and ISR gene targets *Slc7a5*, *Mthfd2*, and *Aars* (*Figure 2C*). However, this analysis was complicated by the finding of substantial co-dependence for expression among these bZIP transcription factors, with knockdown of any one of the C/EBP family members or ATF4 significantly changing expression of at least one other family member. C/EBPδ knockdown, for instance, decreased expression of all genes measured, including ATF4, which was also reflected in loss of ATF4 protein (*Figure 2C, D*). It is worth noting that we were unable to identify reliable antibodies to specific C/EBP family members for use in MEFs. Among C/EBP family members, C/EBPγ has been found in other settings to regulate many of the genes revealed in our analysis to be induced through shared regulation of ATF4 (*Huggins et al., 2015*), and its knockdown significantly decreased expression of the three ATF4 target genes tested without effects on ATF4 protein levels (*Figure 2C, D*). Based on this finding, we knocked down ATF4 or C/EBPγ in wild-type MEFs and stimulated the cells with insulin in the presence or absence of rapamycin to determine whether C/EBPγ impacted the mTORC1 and ATF4-dependent regulation of these genes. Indeed, knockdown of C/EBPγ attenuated the insulin-induced expression of these genes, albeit to a lesser extent than ATF4 knockdown (*Figure 2E*). C/EBPγ knockdown also blocked the ability of insulin to increase ATF4 transcript levels, suggesting that following the induction of ATF4 mRNA translation downstream of mTORC1 (*Ben-Sahra et al., 2016*; *Park et al., 2017*), it stimulates its own expression via ATF4-C/EBPγ heterodimers. Thus, C/EBPγ and likely other ATF4-binding partners of the C/EBP family contribute to the induction of ATF4 gene targets following mTORC1-mediated activation of ATF4.

## mTORC1 signaling induces genes involved in amino acid synthesis, transport, and tRNA charging through ATF4 activation

To validate and expand the findings from the RNA-seq analysis, a NanoString codeset was designed to simultaneously quantify transcripts of genes involved in the enriched processes above (see Materials and methods). As positive and negative controls, respectively, we included the glycolytic targets of HIF1, established previously to be regulated downstream of mTORC1 (*Düvel et al., 2010*), and the mitochondrial tRNA synthetases, not believed to be regulated by ATF4. Using this codeset, we analyzed gene expression in three settings of mTORC1 activation: (1) wild-type MEFs stimulated with insulin in the presence or absence of rapamycin, (2) growth factor-independent activation of mTORC1 via genetic loss of the TSC protein complex in *Tsc2⁻/⁻* MEFs, and (3) *Tsc2⁻/⁻* MEFs with siRNA-mediated knockdown of ATF4 (*Figure 3A*, *Figure 3—source data 1*). ATF4 protein levels were robustly upregulated with either genetic or insulin-stimulated mTORC1 activation in these

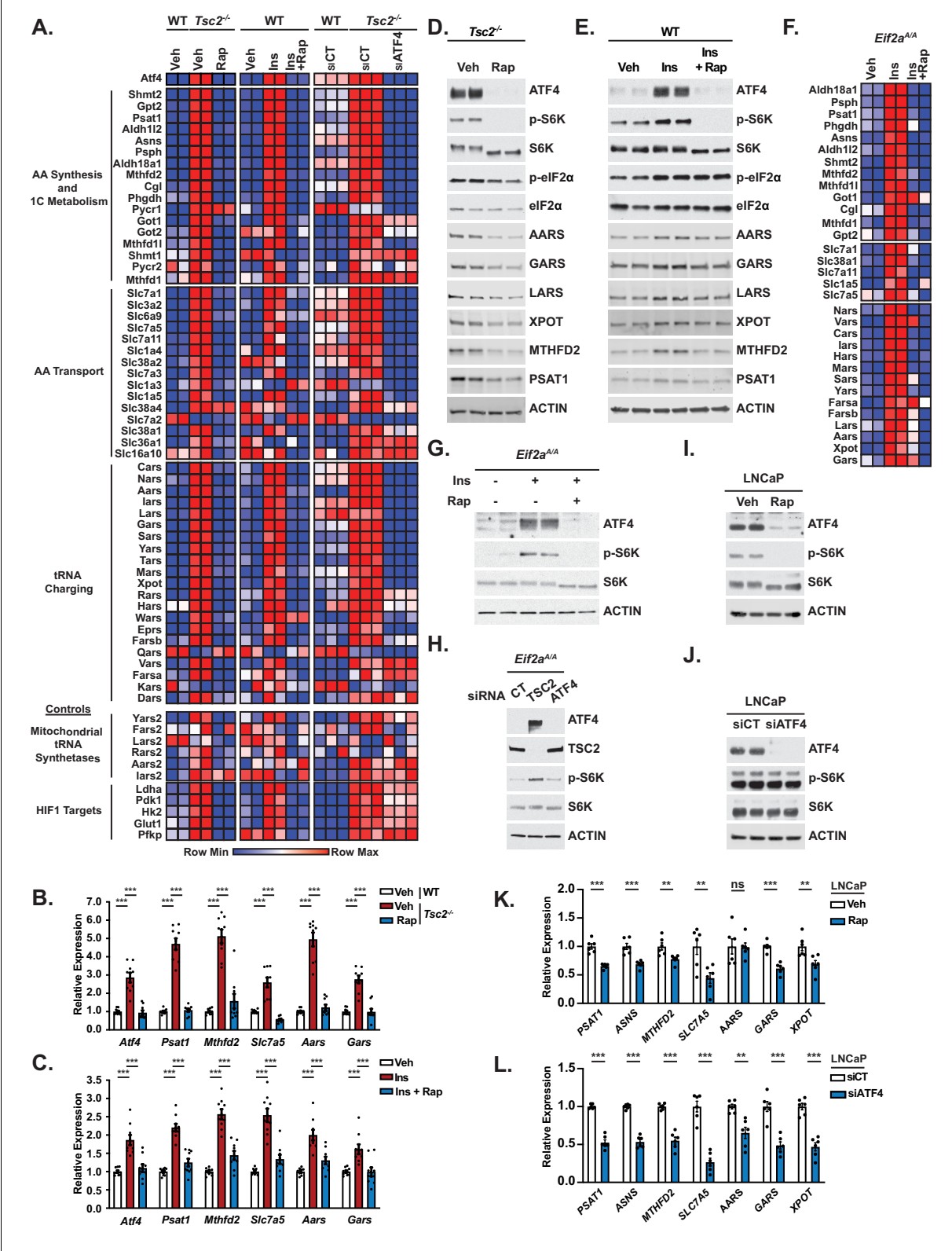

**Figure 3.** Mechanistic target of rapamycin complex 1 (mTORC1) and activating transcription factor 4 (ATF4) regulate genes involved in amino acid synthesis, transport, and tRNA charging. (**A**) Row-normalized heatmaps of NanoString gene expression data are shown from (1) serum-deprived wild-type (WT) or *Tsc2⁻/⁻* mouse embryo fibroblasts (MEFs) treated (16 hr) with vehicle (Veh) or rapamycin (20 nM, Rap) (n = 2), (2) serum-deprived WT MEFs treated with insulin (500 nM, 16 hr, Ins) following 30 min pretreatment with vehicle or rapamycin (20 nM) (n = 2), and (3) WT and *Tsc2⁻/⁻* MEFs transfected
*Figure 3 continued on next page*

*Figure 3 continued*

with *Atf4* siRNAs or non-targeting controls (siCT) and serum-deprived for 16 hr (n = 3). Genes are grouped by functional category and ranked in order of most significantly decreased with ATF4 knockdown for each group. Heatmap values are provided in *Figure 3—source data 1*, and effects on ATF4 protein levels and mTORC1 signaling for each condition are shown by immunoblot in *Figure 3—figure supplement 1A*. (B, C) qPCR analysis of the indicated transcripts in WT and *Tsc2*$^{-/-}$ MEFs (B) or WT MEFs stimulated with insulin in the presence or absence of rapamycin (C) as in (A). Expression relative to vehicle-treated, unstimulated WT cells is graphed as mean ± SEM from three independent experiments, each with three biological replicates (n = 9). Effects of ATF4 knockdown are shown in *Figure 3—figure supplement 1B*. (D, E) Representative immunoblots of *Tsc2*$^{-/-}$ MEFs treated with vehicle or rapamycin as in (A) or serum-deprived WT MEFs stimulated with insulin (500 nM, 24 hr) following 30 min pretreatment with vehicle or rapamycin (20 nM), with biological duplicates shown for each condition. Quantification provided in *Figure 3—figure supplement 1C, D*. (F) Row-normalized heatmaps of NanoString gene expression data for transcripts in the functional groups from (A) found to be significantly (p<0.05) induced in *eIF2α*$^{A/A}$ MEFs treated with insulin (500 nM, 16 hr) following 30 min pretreatment with vehicle or rapamycin (20 nM) (n = 2). Genes are ranked by category in order of most significantly increased with insulin for each group. The heatmap values are provided in *Figure 3—source data 1*. Immunoblots validating that these cells are defective in the integrated stress response and qPCR validation of representative genes are provided in *Figure 3—figure supplement 1E, F*. (G) Representative immunoblot of cells treated as in (F), with biological duplicates shown for each condition. (H) Representative immunoblot of *eIF2α*$^{A/A}$ MEFs transfected with siRNAs targeting *Atf4, Tsc2,* or non-targeting controls (CT) prior to serum starvation for 16 hr. (I, J) Representative immunoblot of serum-deprived LNCaP cells treated with vehicle or rapamycin (20 nM, 16 hr) (I) or *Atf4* siRNAs versus non-targeting controls (siCT) (J), with biological duplicates shown for each. (K, L) qPCR analysis of the indicated transcripts in LNCaP cells serum-starved in the presence of vehicle or rapamycin (20 nM, 16 hr) (K) or transfected with *ATF4* siRNAs or non-targeting controls (siCT) and serum-starved for 16 hr (L). Expression relative to vehicle-treated cells is graphed as mean ± SEM from two independent experiments, with three biological replicates each (n = 6). Immunoblots and qPCR analysis for PC3 cells treated as in (I–L) are provided in *Figure 3—figure supplement 1G–J*, and effects of c-Myc knockdown on representative gene targets are shown in *Figure 3—figure supplement 1K, L*. *p<0.05, **p<0.01, ***p<0.001, ns = not significant. One-way analysis of variance with Holm–Sidak method for multiple comparisons was used to determine statistical significance for (B, C). Unpaired two-tailed *t*-test was used to determine statistical significance for (F, K, L). (D, E, G, H, I, J) are representative of at least two independent experiments. The online version of this article includes the following source data and figure supplement(s) for figure 3:

**Source data 1.** Nanonstring data supporting *Figure 3*.
**Figure supplement 1.** Supplmental data supporting *Figure 3*.

settings, with both rapamycin and ATF4-targeting siRNAs blocking this induction (*Figure 3—figure supplement 1A*). Interestingly, the majority of transcripts analyzed in the functional categories that encode enzymes of NEAA synthesis, one-carbon metabolism, amino acid transporters, and cytosolic aminoacyl tRNA synthetases (and *Xpot*) were increased with mTORC1 activation in a manner sensitive to both rapamycin and siRNA-mediated knockdown of ATF4. However, the HIF1 targets of glycolysis were mTORC1-regulated but independent of ATF4, and transcripts encoding the mitochondrial tRNA synthetases were reproducibly regulated by neither mTORC1- nor ATF4. We further confirmed the mTORC1- and ATF4-mediated regulation of a representative subset of these transcripts via qPCR (*Figure 3B, C*, *Figure 3—figure supplement 1B*). Consistent with previous studies (*Ben-Sahra et al., 2016*; *Park et al., 2017*), both ATF4 transcript and protein levels were induced by mTORC1 signaling in these settings (*Figure 3B–E*). Transcriptional changes in ATF4 gene targets were reflected in corresponding changes in the abundance of representative protein products, with varying degrees of rapamycin sensitivity, likely reflecting inherent differences in the turnover rates of these proteins (*Figure 3D, E*, *Figure 3—figure supplement 1C, D*). We next wanted to confirm that these mTORC1- and ATF4-induced changes were independent of the ISR. While we have shown previously that mTORC1 regulates ATF4 in a manner that is independent of eIF2α-S51 phosphorylation (*Ben-Sahra et al., 2016*), chronic activation of mTORC1 upon loss of TSC2 is known to cause a basal increase in ER stress and activation of the ISR (*Ozcan et al., 2008*). Therefore, we utilized MEFs with endogenous, homozygous knock-in of the *Eif2a-S51A* mutation (*Eif2a*$^{A/A}$), which fail to induce ATF4 downstream of cellular stress (*Figure 3—figure supplement 1E*; *Scheuner et al., 2001*). Consistent with mTORC1-dependent, ISR-independent regulation, insulin increased ATF4 protein levels and expression of its gene targets involved in amino acid synthesis and one-carbon metabolism, amino acid transporters, and aminoacyl tRNA synthetases in a rapamycin-sensitive manner in these cells, as shown by NanoString analysis and confirmed for a subset of genes by qPCR (*Figure 3F, G*, *Figure 3—figure supplement 1F*). Notably, like insulin stimulation, genetic activation of mTORC1 via siRNA-mediated knockdown of *Tsc2* in the *eIF2α*$^{A/A}$ MEFs also increased ATF4 protein levels, further confirming that this regulation can occur independent of the ISR (*Figure 3H*). Growth factor-independent activation of mTORC1 also occurs upon loss of the PTEN tumor suppressor, and rapamycin was found to decrease ATF4 protein levels and ATF4-

dependent expression of representative gene targets in established PTEN-deficient prostate cancer cells, LNCaP and PC3 (*Figure 3I–L*, *Figure 3—figure supplement 1G–J*).

The transcription factor c-MYC can be activated downstream of mTORC1 in some settings and has previously been shown to regulate many of the same ATF4 target genes encoding the enzymes of amino acid synthesis and transport, as well as aminoacyl tRNA synthetases (*Csibi et al., 2014*; *Stine et al., 2015*; *Wall et al., 2008*; *Zirin et al., 2019*). A survey of expression for 11 of such genes in *Tsc2$^{-/-}$* cells finds that the majority are not significantly affected by siRNA-mediated knockdown of c-MYC, whereas a few others are modestly but significantly decreased (e.g., *Gars*, *Slc1a5*, *Psat1*), albeit to a lesser degree than with siRNA-mediated knockdown of ATF4 (*Figure 3—figure supplement 1K, L*).

To determine whether ATF4 activation is both necessary and sufficient for mTORC1-mediated regulation of these gene targets related to amino acid acquisition and utilization, we knocked out *Atf4* using CRISPR/Cas9 in *Tsc2$^{-/-}$* MEFs and confirmed biallelic disruption (*Figure 4A*). Protein levels of identified ATF4 targets were decreased in *Tsc2$^{-/-}$ Atf4$^{-/-}$* MEFs and fully rescued with expression of wild-type ATF4 but not a DNA-binding domain mutant (ATF4$^{DBD}$) (*Figure 4B*). As mTORC1 regulates ATF4 translation through a mechanism requiring its 5′-UTR (*Ben-Sahra et al., 2016*; *Park et al., 2017*), the stably rescued cell lines, which express an ATF4 cDNA lacking the 5′-UTR, exhibit protein expression of ATF4 and its encoded gene targets that are resistant to rapamycin (*Figure 4C*, *Figure 4—figure supplement 1*). The expression of select ATF4 target transcripts was markedly decreased in *Tsc2$^{-/-}$ Atf4$^{-/-}$* MEFs to a similar extent to that measured in *Atf4* wild-type cells treated with rapamycin (*Figure 4D*). These transcript levels were rescued with the expression of wild-type ATF4, but not ATF4$^{DBD}$, in a manner that was completely or partially rapamycin resistant (*Figure 4D*). These collective data show that mTORC1 signaling drives the expression of genes involved in tRNA export and charging, amino acid uptake, and NEAA synthesis through its downstream regulation of the ATF4 transcription factor.

## Activation of ATF4 contributes to the induction of protein synthesis downstream of mTORC1

mTORC1 induces protein synthesis through multiple downstream targets (*Valvezan and Manning, 2019*). Given that the major mTORC1-regulated ATF4 target genes identified above are involved in amino acid uptake, synthesis, and tRNA charging, we hypothesized that ATF4 induction through mTORC1 signaling might contribute to the canonical increase in protein synthesis upon mTORC1 activation. Relative rates of protein synthesis were measured via [$^{35}$S]-methionine incorporation into newly synthesized proteins in *Tsc2$^{+/+}$* and *Tsc2$^{-/-}$* cells treated with control siRNAs or *Tsc2$^{-/-}$* cells treated with siRNAs targeting ATF4 or Rheb, the small GTPase target of TSC2 that is an essential upstream activator of mTORC1. siRNA-mediated knockdown of either ATF4 or Rheb substantially decreased ATF4 protein levels in *Tsc2$^{-/-}$* cells (*Figure 5A*). Importantly, the elevated rate of protein synthesis in *Tsc2$^{-/-}$* MEFs was decreased with ATF4 knockdown to a similar extent to that observed with Rheb knockdown (*Figure 5A, B*). No change in mTORC1 signaling or phosphorylation of eIF2α was observed with ATF4 knockdown in this setting (*Figure 5—figure supplement 1A*). Protein synthesis was also measured in the *Tsc2$^{-/-}$ Atf4$^{-/-}$* cell lines described above. Notably, the cells lacking ATF4 exhibited increased uptake of [$^{35}$S]-methionine relative to parental cells and those reconstituted with ATF4 (*Figure 5—figure supplement 1B*). Despite this unexplained difference in methionine uptake, *Atf4* knockout cells exhibited a reduced rate of protein synthesis that was similar to the parental lines treated with rapamycin (*Figure 5C, D*). However, rapamycin treatment further reduced protein synthesis in the *Atf4* knockout cells. Importantly, cells reconstituted with the rapamycin-resistant ATF4 cDNA exhibited rescued protein synthesis, surpassing that observed in cells with endogenous ATF4, but this enhanced protein synthesis was still significantly reduced with rapamycin. Like *Tsc2$^{-/-}$* cells, wild-type cells cultured in the presence of growth factors exhibited reduced protein synthesis upon deletion of *Atf4*, again with a reduction similar to that from rapamycin treatment (*Figure 5—figure supplement 1C, D*). Together, these data suggest that ATF4 induction downstream of mTORC1 is necessary but not sufficient for mTORC1-regulated protein synthesis, consistent with the multiple mechanisms through which mTORC1 controls this key anabolic process.

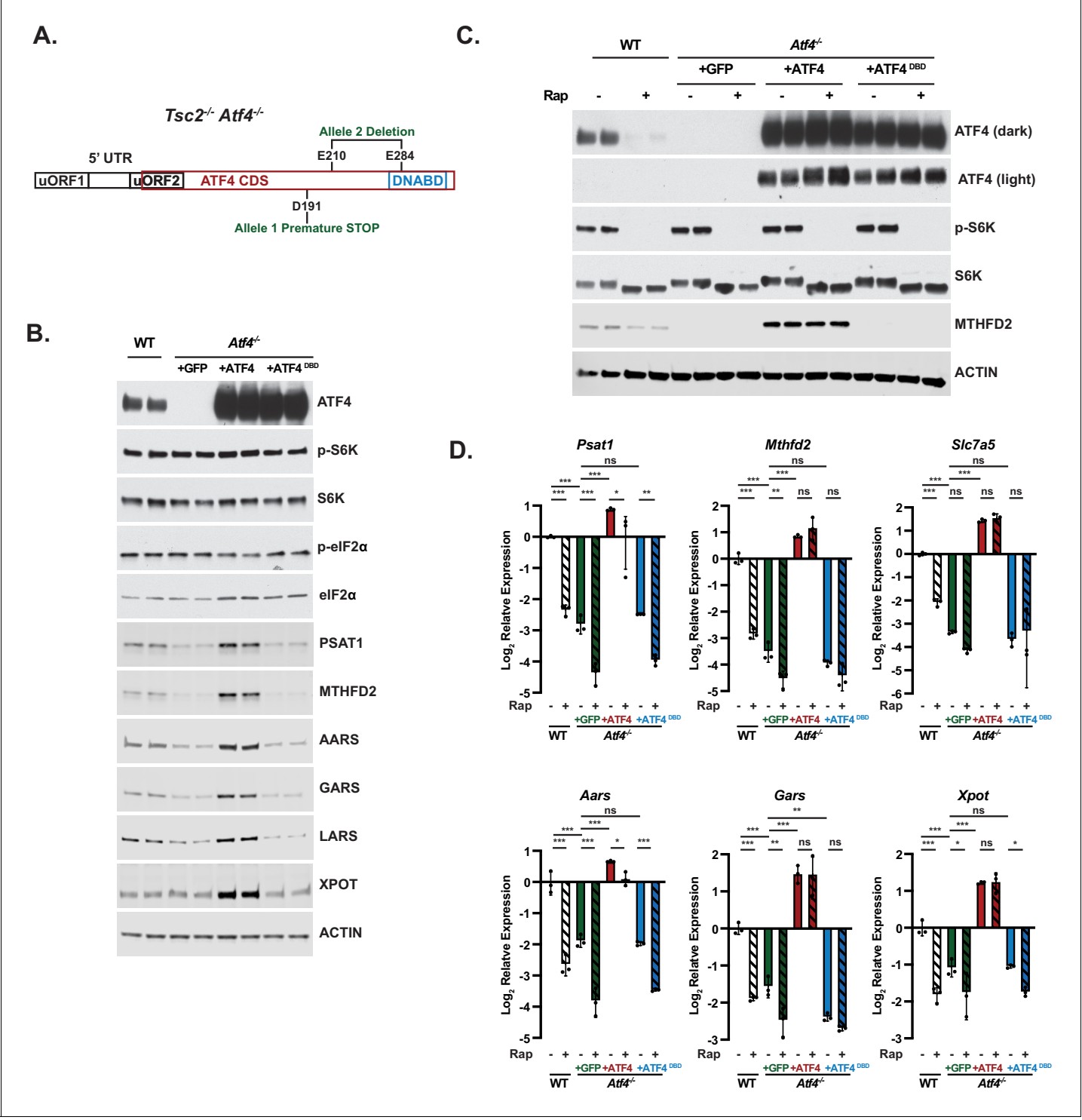

**Figure 4.** Use of activating transcription factor 4 (ATF4) knockout cells and a rapamycin-resistant ATF4 to validate ATF4 targets regulated by mechanistic target of rapamycin complex 1 signaling. (A) Schematic of ATF4 transcript, including upstream open reading frames (uORFs), coding sequence (CDS), and DNA-binding domain (DNABD), highlighting location of CRISPRn guides biallelic location of ATF4 mutations generated in *Tsc2⁻/⁻* mouse embryo fibroblasts (MEFs). (B, C) Representative immunoblots of serum-deprived *Tsc2⁻/⁻* (wild-type [WT]) MEFs or *Tsc2⁻/⁻ Atf4⁻/⁻* MEFs with stable expression of cDNAs encoding GFP, ATF4 lacking its 5′-UTR, or a DNABD mutant (DBD) of this ATF4 left untreated (B) or treated with vehicle or rapamycin (20 nM, 16 hr) (C), with biological duplicates shown for each condition. Immunoblots of proteins encoded by ATF4 gene targets in these cells are provided in *Figure 4—figure supplement 1*. (D) qPCR analysis of the indicated transcripts from cells treated as in (C). Expression relative to WT vehicle-treated cells is graphed as the $\log_2$ mean ± SD from a representative experiment with three biological replicates (n = 3). *p<0.05, **p<0.01,

*Figure 4 continued on next page*

*Figure 4 continued*

***p<0.001, ns = not significant, calculated via one-way analysis of variance with Holm–Sidak method for multiple comparisons. (**B–D**) are representative of at least two independent experiments.

The online version of this article includes the following figure supplement(s) for figure 4:

**Figure supplement 1.** Supplemental data supporting *Figure 4*.

## mTORC1 regulates cystine uptake through ATF4

Among the 61 shared mTORC1- and ISR-induced ATF4 gene targets identified, the cystine-glutamate antiporter *Slc7a11* was the gene with the highest fold induction by RNA-seq analysis upon insulin treatment (*Figure 1F*). SLC7A11 (also known as xCT) associates with SLC3A2 (CD98) at the plasma membrane and serves as the primary transporter of cystine, the oxidized form of cysteine and predominant cysteine species in both plasma and cell culture media, whereas reduced cysteine is transported through neutral amino acid systems (*Figure 6A*; *Bannai and Kitamura, 1980*; *Conrad and Sato, 2012*). Transcript levels of *Slc7a11* were sensitive to rapamycin in *Tsc2$^{-/-}$* MEFs and greatly decreased with ATF4 knockout (*Figure 6B*). Upon reconstitution with rapamycin-resistant ATF4, expression of *Slc7a11* was rescued and no longer sensitive to rapamycin, while the ATF4$^{DBD}$ mutant was unable to restore *Slc7a11* transcript levels. A similar pattern of mTORC1- and ATF4-regulated expression was measured for *Slc3a2* (*Figure 6C*). SLC7A11 protein, detected using an antibody validated with siRNA knockdown (*Figure 6—figure supplement 1A*), decreased in *Tsc2$^{-/-}$* MEFs treated with mTOR inhibitors and were increased in wild-type MEFs stimulated with insulin in an mTOR-dependent manner (*Figure 6D, E*, *Figure 6—figure supplement 1B, C*). *SLC7A11* transcript levels were also decreased with both ATF4 knockdown and rapamycin in the PTEN-deficient cancer cell lines LNCaP and PC3, although *SLC7A11* expression was relatively more resistant to rapamycin in PC3 cells (*Figure 6—figure supplement 1D, E*). SLC7A11 protein levels likewise decreased in LNCaP and PC3 cells treated with mTOR inhibitors, without significant changes to the *SLC3A2* gene product CD98 (*Figure 6F*, *Figure 6—figure supplement 1F–H*). These data confirm and extend the findings from RNA-seq and NanoString analyses (*Figure 1F*, *Figure 3A, F*) and demonstrate that ATF4 is both necessary and sufficient for the mTORC1-mediated induction of *Slc7a11* expression.

ATF4 is known to be important for the uptake and synthesis of NEAAs (*Harding et al., 2003*). In agreement with this, we observed that *Tsc2$^{-/-}$ Atf4$^{-/-}$* cells fail to proliferate in Dulbecco's Modified Eagle's Medium (DMEM), which only contains a subset of NEAAs, while addback of wild-type ATF4, but not ATF4$^{DBD}$, restored proliferation (*Figure 6G*). Supplementation of DMEM with a mixture of all NEAAs, including cysteine, allowed the *Tsc2$^{-/-}$ Atf4$^{-/-}$* MEFs to proliferate at the same rate as the ATF4-reconstituted cells, while NEAAs lacking cysteine completely failed to support proliferation of these cells (*Figure 6H*). Furthermore, supplementation with excess reduced cysteine alone, but not equimolar concentrations of oxidized cysteine in the form of cystine, was able to significantly increase proliferation of the *Tsc2$^{-/-}$ Atf4$^{-/-}$* MEFs, albeit to a lesser extent than NEAAs plus cysteine. The majority of these cells die after 72 hr in DMEM, and exogenous expression of either ATF4 or SLC7A11 restores their survival (*Figure 6I*). Taken together, these data indicate that a defect in the acquisition of cysteine, which normally occurs through SLC7A11-mediated uptake of cystine, underlies the inability of *Tsc2$^{-/-}$ Atf4$^{-/-}$* cells to proliferate or survive in DMEM and suggest a key role for mTORC1 signaling in controlling cystine uptake through ATF4.

To directly test whether mTORC1 influences cystine uptake, we employed both genetic (*Tsc2* loss) and physiological (insulin stimulation) activation of mTORC1, measuring [$^{14}$C]-cystine uptake in the presence or absence of rapamycin or the xCT inhibitor erastin (*Figure 6A*; *Yang and Stockwell, 2008*). Both rapamycin and erastin significantly decreased [$^{14}$C]-cystine uptake into *Tsc2$^{-/-}$* MEFs (*Figure 6J*). In wild-type MEFs, insulin stimulated an increase in cystine uptake that was inhibited with rapamycin, and this mTORC1-regulated cystine transport was completely lost with ATF4 knockout, reduced to levels of erastin-treated cells (*Figure 6K*). Additionally, *Tsc2$^{-/-}$ Atf4$^{-/-}$* MEFs showed a decrease in cystine uptake when compared to parental *Tsc2$^{-/-}$* MEFs, which could be rescued with re-expression of ATF4 (*Figure 6L*). However, cystine consumption in cells reconstituted with rapamycin-resistant ATF4 was still significantly sensitive to rapamycin treatment, suggesting the existence of additional, ATF4-independent mechanisms influencing the transport or cellular incorporation of

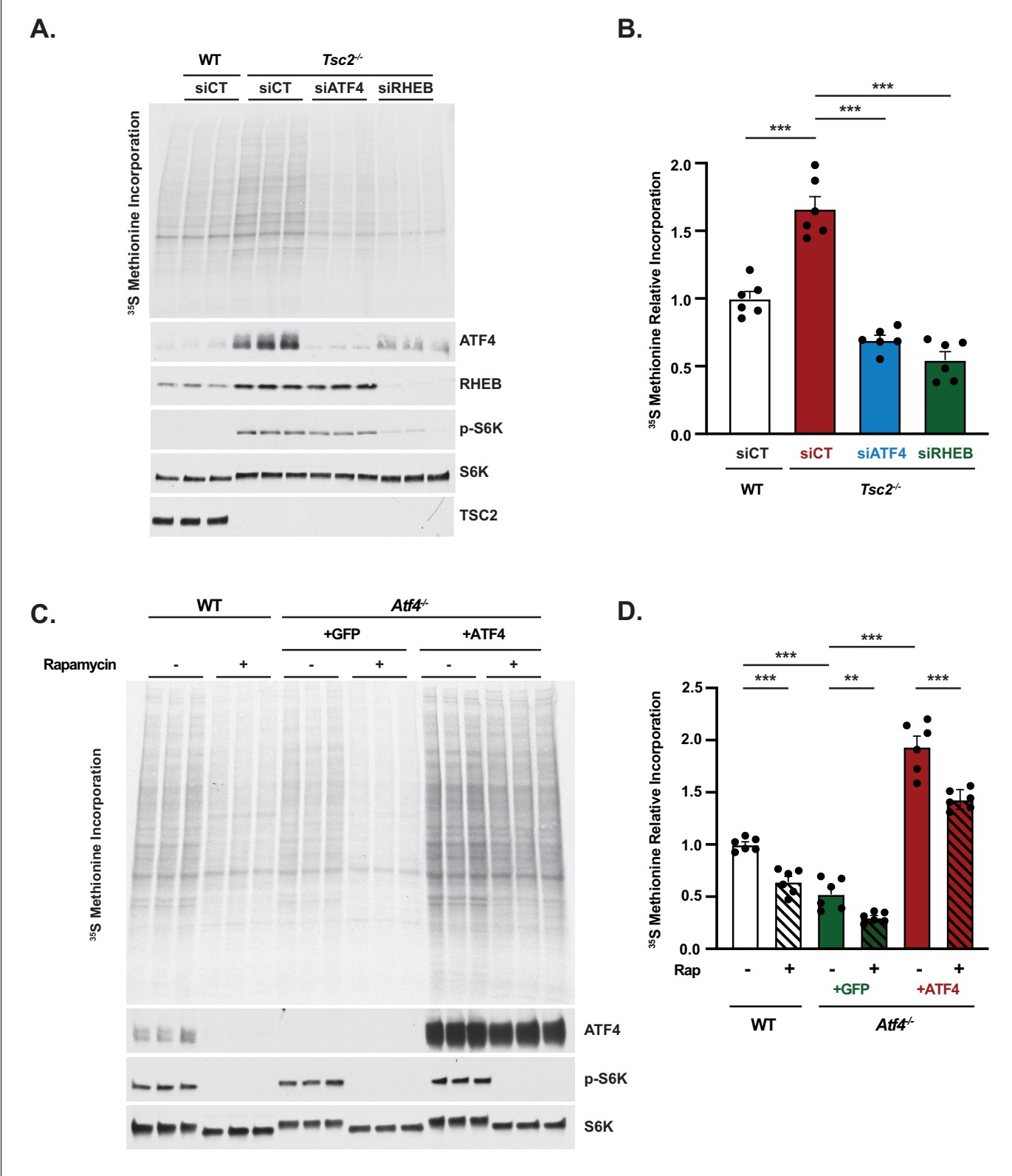

**Figure 5.** Activation of activating transcription factor 4 (ATF4) contributes to the induction of protein synthesis downstream of mechanistic target of rapamycin complex 1. (A, B) Representative autoradiogram and immunoblot of wild-type (WT) and *Tsc2⁻/⁻* mouse embryo fibroblasts (MEFs) transfected with siRNAs targeting *Atf4* or *Rheb1* and *Rhebl1* or non-targeting controls (siCT) and serum-deprived for 16 hr with a pulse label of [³⁵S]-methionine for the final 20 min (A) and quantified in (B). Biological triplicates from a representative experiment are shown in (A). (B) is graphed as mean ± SEM from

*Figure 5 continued on next page*

*Figure 5 continued*

two independent experiments, each with three biological replicates (n = 6). Lack of effect of ATF4 knockdown on eIF2α phosphorylation is shown in *Figure 5—figure supplement 1A*. (C, D) Representative autoradiogram and immunoblot of serum-deprived *Tsc2⁻/⁻* MEFs (WT) or *Tsc2⁻/⁻ Atf4⁻/⁻* MEFs with stable expression of cDNAs encoding GFP or ATF4 lacking its 5'-UTR treated with vehicle or rapamycin (20 nM, 16 hr) with a pulse label of [$^{35}$S]-methionine for the final 20 min (C) and quantified in (D). Biological triplicates from a representative experiment are shown in (C). (D) is graphed as mean ± SEM from two independent experiments, each with three biological replicates (n = 6). Measurement of methionine uptake in these cells is provided in *Figure 5—figure supplement 1B*, and effects of ATF4 knockout on protein synthesis in growth factor-stimulated WT MEFs are shown in *Figure 5—figure supplement 1C, D*. (B,D) *p<0.05, **p<0.01, ***p<0.001, ns = not significant, calculated via one-way analysis of variance with Holm–Sidak method for multiple comparisons.

The online version of this article includes the following figure supplement(s) for figure 5:

**Figure supplement 1.** Supplemental data supporting *Figure 5*.

cystine downstream of mTORC1. As mTORC2 has been previously suggested to directly regulate xCT (*Gu et al., 2017*), we utilized *Rictor⁻/⁻* MEFs, which lack mTORC2 activity, to determine whether mTORC2 was contributing to the decreased cystine uptake observed upon treatment with mTOR inhibitors. While *Rictor⁻/⁻* MEFs displayed increased uptake of cystine relative to their wild-type counterparts, cystine uptake, as well as ATF4 protein levels, was sensitive to rapamycin and Torin1 in both cell lines (*Figure 6—figure supplement 1I, J*). Thus, mTORC1 promotes cellular cystine uptake, at least in part, through the activation of ATF4 and induction of *Slc7a11* expression, which supports cell proliferation and survival.

## mTORC1 regulates glutathione levels through ATF4-mediated induction of *Slc7a11*

Cysteine, generally acquired through cystine uptake and reduction, is an essential component of the tripeptide glutathione (*Figure 6A*), the most abundant antioxidant in cells (*Meister, 1983*). We hypothesized that the regulation of cystine uptake through mTORC1 and ATF4 might influence cellular glutathione content. Indeed, mTORC1 inhibition with rapamycin or Torin1 significantly decreased total glutathione levels in *Tsc2⁻/⁻* MEFs, albeit less than buthionine-sulfoximine (BSO), a direct inhibitor of glutathione synthesis (*Figure 7A*; *Griffith and Meister, 1979*). Similar to BSO treatment, mTOR inhibitors decreased both reduced (GSH) and oxidized (GSSG) forms of glutathione to the same degree, indicating effects on total glutathione abundance rather than its redox state (*Figure 7—figure supplement 1A*). Stable reconstitution of *Tsc2⁻/⁻* MEFs with TSC2 also decreased total glutathione levels (*Figure 7B*). To examine this response in vivo, we employed a mouse model of tuberous sclerosis complex involving xenograft tumors derived from the rat *TSC2⁻/⁻* tumor cell line ELT3 (*Hodges et al., 2002*). To avoid major differences in tumor size from the treatments, we treated tumor-bearing mice for just 5 days with either vehicle or rapamycin, prior to harvesting tumors for immunoblot analysis and metabolite profiling. Importantly, we found that rapamycin treatment strongly decreased ATF4 protein levels in these tumors with a concomitant decrease in glutathione levels, measured by LC-MS in tumor metabolite extracts (*Figure 7C, D*). An analysis of published metabolomics data (*Tang et al., 2019*) also revealed that rapamycin treatment significantly decreased glutathione levels in human TSC2-deficient angiomyolipoma cells (*Figure 7—figure supplement 1B*). Likewise, inhibition of mTORC1 signaling with rapamycin or Torin1 in LNCaP and PC3 cells resulted in a significant decrease in total glutathione levels, although the degree of decrease varied between the two cell lines (*Figure 7E*), perhaps reflecting the above finding that SLC7A11 expression is more resistant to mTOR inhibitors in PC3 cells (*Figure 6—figure supplement 1D–H*).

To determine the role of ATF4 and SLC7A11-dependent cystine uptake in glutathione synthesis downstream of mTORC1 signaling, we compared *Tsc2⁻/⁻* MEFs with or without *Atf4* knockout. Total glutathione levels were greatly decreased in *Tsc2⁻/⁻ Atf4⁻/⁻* MEFs, and exogenous expression of ATF4 or SLC7A11, but not ATF4$^{DBD}$, was able to restore glutathione levels to these cells (*Figure 7F*). Supplementation with all NEAAs or just cysteine, transported through neutral amino acid systems, but not equimolar concentrations of cystine, transported through SLC7A11, also rescued total glutathione levels, as measured by either enzymatic assay or LC-MS (*Figure 7F*, *Figure 7—figure supplement 1C*). Furthermore, insulin stimulated an increase in glutathione levels in wild-type MEFs in a manner completely sensitive to rapamycin, an effect ablated in *Atf4* knockout cells, which had a very

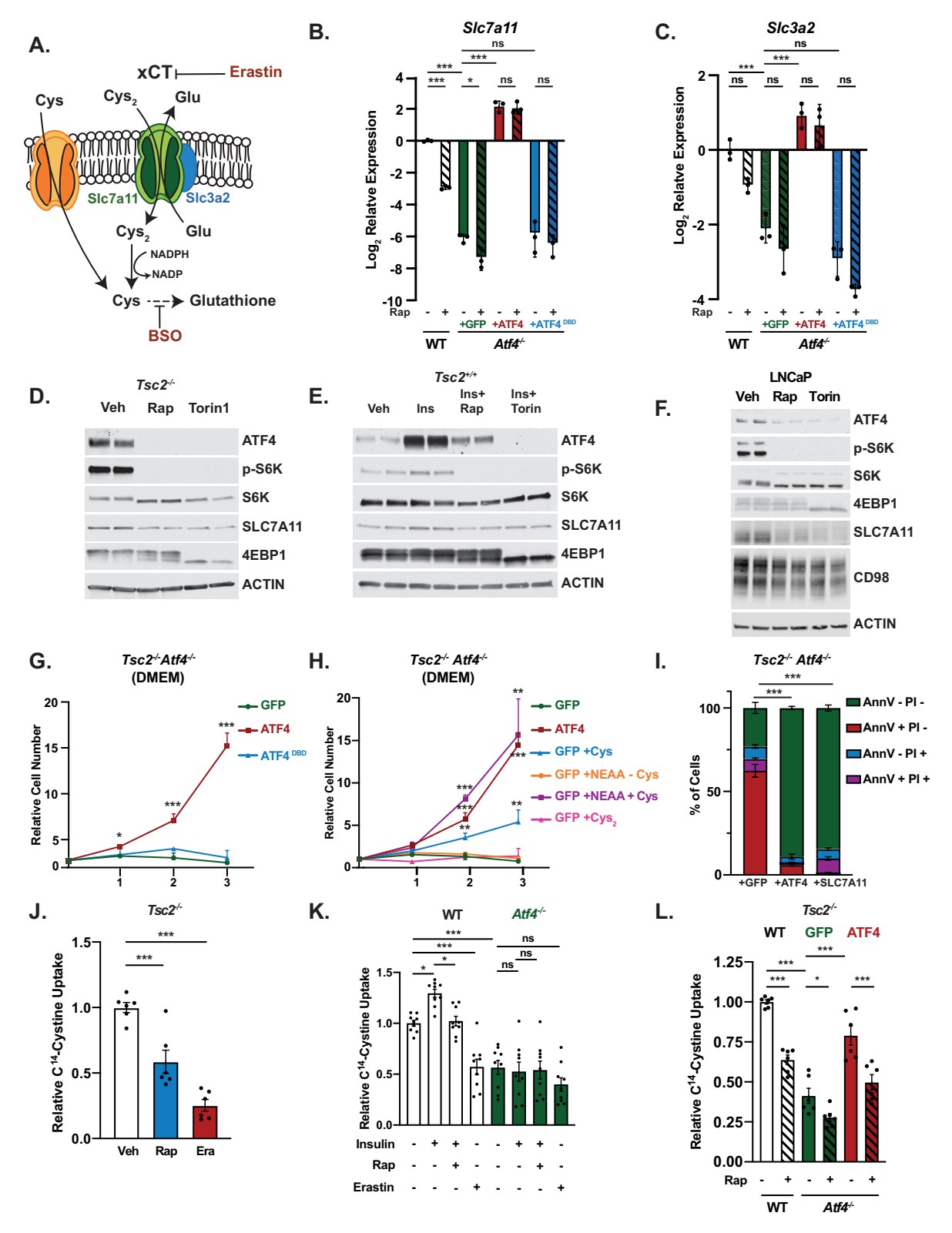

**Figure 6.** Mechanistic target of rapamycin complex 1 (mTORC1) regulates cystine uptake through activating transcription factor 4 (ATF4) and its target SLC7A11. (**A**) Schematic of transporter xCT, encoded by *Slc7a11*, which heterodimerizes with SLC3A2 to serve as a cystine (Cys₂)/glutamate anti-porter. Cystine is reduced to cysteine (Cys), which is essential for glutathione synthesis. Cysteine transport is mediated by neutral amino acid trasporters distinct from xCT. The targets of erastin and buthionine-sulfoximine, two compounds used in this study, are also depicted. (**B, C**) qPCR analysis of

*Figure 6 continued on next page*

*Figure 6 continued*

*Slc7a11* (**B**) or *Slc3a2* (**C**) in serum-deprived *Tsc2*$^{-/-}$ mouse embryo fibroblasts (MEFs) (wild-type [WT]) and *Tsc2*$^{-/-}$ *Atf4*$^{-/-}$ MEFs with stable expression of cDNAs encoding GFP (control), ATF4 lacking its 5′-UTR, or a DNABD mutant (DBD) of this ATF4 treated with vehicle or rapamycin (20 nM, 16 hr). Expression relative to WT vehicle-treated cells is graphed as the log$_2$ mean ± SD from a representative experiment with three biological replicates (n = 3). (**D–F**) Representative immunoblots of serum-deprived *Tsc2*$^{-/-}$ MEFs (**D**), insulin-stimulated (500 nM, 24 hr) WT MEFs (**E**), or serum-deprived LNCaP cells (**F**) treated 24 hr (**D, E**) or 16 hr (**F**) with vehicle, rapamycin (20 nM), or Torin1 (250 nM), shown with biological duplicates. The SLC7A11 antibody is validated for use in MEFs in *Figure 6—figure supplement 1A*, corresponding immunoblots quantified in *Figure 6—figure supplement 1B, C, F*. Effects of ATF4 knockdown and rapamycin on *SLC7A11* transcripts in LNCaP and PC3 cells are shown in *Figure 6—figure supplement 1D, E*, and corresponding immunoblots and protein quantification are provided in *Figure 6—figure supplement 1G, H*. (**G, H**) Representative growth curves of *Tsc2*$^{-/-}$ *Atf4*$^{-/-}$ MEFs with stable expression of GFP, ATF4, or ATF4$^{DBD}$ grown in 10% dialyzed fetal bovine serum (FBS) with Dulbecco's Modified Eagle's Medium (DMEM) (**G**) or DMEM supplemented with cysteine alone (Cys, 1 mM), nonessential amino acids (100 μM each) lacking cysteine (NEAA-Cys), or nonessential amino acids plus either cysteine (1 mM, NEAA+Cys), or cystine (0.5 mM, NEAA+Cys$_2$) (**H**). Mean cell numbers ± SD relative to day 0 are graphed from three biological replicates (n = 3). (**I**) Cell death of *Tsc2*$^{-/-}$ *Atf4*$^{-/-}$ MEFs with stable expression of cDNAs encoding GFP, ATF4, or SLC7A11 cultured in DMEM with 10% dialyzed FBS was quantified by annexin V and propidium iodide (PI) staining after 72 hr and graphed as the mean percentage of total cells ± SD from three biological replicates (n = 3). (**J**) Cystine uptake in serum-deprived *Tsc2*$^{-/-}$ MEFs treated with vehicle, rapamycin (20 nM), or erastin (10 μM) for 16 hr is graphed as the mean ± SEM radiolabel incorporation from C$^{14}$-cystine over the final 10 min relative to vehicle-treated cells from two independent experiments, with three biological replicates each (n = 6). The effect of mTOR inhibitors on cystine uptake in littermate-derived *Rictor*$^{+/+}$ and *Rictor*$^{-/-}$ MEFs, and corresponding immunoblots, is shown in *Figure 6—figure supplement 1I, J*. (**K**) Cystine uptake in serum-deprived WT and *Atf4*$^{-/-}$ MEFs pretreated 30 min with vehicle or rapamycin (20 nM) prior to insulin stimulation (500 nM, 24 hr) or treated with erastin (10 μM, 30 min) was assayed and graphed as in (**J**) relative to vehicle-treated WT cells with data from three independent experiments, with three biological replicates each (n = 9). (**L**) Cystine uptake in serum-deprived *Tsc2*$^{-/-}$ MEFs (WT) and *Tsc2*$^{-/-}$ *Atf4*$^{-/-}$ MEFs with stable expression of cDNAs encoding GFP or ATF4 treated with vehicle or rapamycin (20 nM) for 16 hr was assayed and graphed as in (**J**) relative to vehicle-treated WT cells with data from two independent experiments, with three biological replicates each (n = 6). (**B–I**) are representative of at least two independent experiments. *p<0.05, **p<0.01, ***p<0.001, ns = not significant, calculated in (**B, C, I, J, K, L**) via one-way analysis of variance with Holm–Sidak method for multiple comparisons and in (**G, H**) via unpaired two-tailed *t*-test. For (**I**), the sum of annexin V+/PI-, annexin V-/PI+, and annexin V+/PI+ populations were used for comparisons to the annexin V-/P- population.

The online version of this article includes the following figure supplement(s) for figure 6:

**Figure supplement 1.** Supplemental data supporting *Figure 6*.

---

low abundance of glutathione (*Figure 7G*). Likewise, the rapamycin-sensitive nature of glutathione in *Tsc2*$^{-/-}$ MEFs was completely lost in *Tsc2*$^{-/-}$ *Atf4*$^{-/-}$ MEFs (*Figure 7H*). Glutathione levels were restored to these cells upon exogenous expression of ATF4, but not the ATF4$^{DBD}$ mutant. However, glutathione was still significantly reduced with rapamycin treatment in cells expressing the rapamycin-resistant ATF4, suggesting possible ATF4-independent mechanisms also contributing to this regulation (*Lam et al., 2017*). As one possible contributing factor, we found that the transcript encoding both the catalytic (GCLC) and regulatory (GCLM) subunits of glutamate-cysteine ligase, the first enzyme of glutathione synthesis, was sensitive rapamycin, in a manner unaffected by ATF4 knockdown (*Figure 7—figure supplement 1D, E*). We also found that GCLC and GCLM protein levels could be modestly induced by insulin through mTORC1 signaling in both wild-type and ATF4 knockout cells, but their protein abundance was unaffected by rapamycin in *Tsc2*$^{-/-}$ MEFs (*Figure 7—figure supplement 1F, G*). Thus, the mechanism underlying the apparent ATF4-independent effects of mTORC1 signaling on glutathione levels remains unknown. These collective data show that mTORC1 signaling induces glutathione synthesis, at least in part, through the activation of ATF4 and SLC7A11-dependent cystine uptake.

## Discussion

Our findings expand the functional repertoire of mTORC1 signaling as it relates to the control of anabolic processes and cellular metabolism through its noncanonical activation of ATF4. Importantly, less than 10% of stress-responsive, ATF4-dependent targets were found to be significantly stimulated through the mTORC1-mediated activation of ATF4 in response to insulin. Among others, we found that genes involved in amino acid biosynthesis, transport, and tRNA charging were induced by mTORC1-ATF4 signaling, many to a comparable level to that of ER-stress induction with tunicamycin. While the molecular nature of this selective induction remains unknown, our data suggest that the 61 ATF4-dependent genes shared in their induction between mTORC1 signaling and the ISR represent targets most highly responsive to increases in ATF4 levels. Since mTORC1 signaling

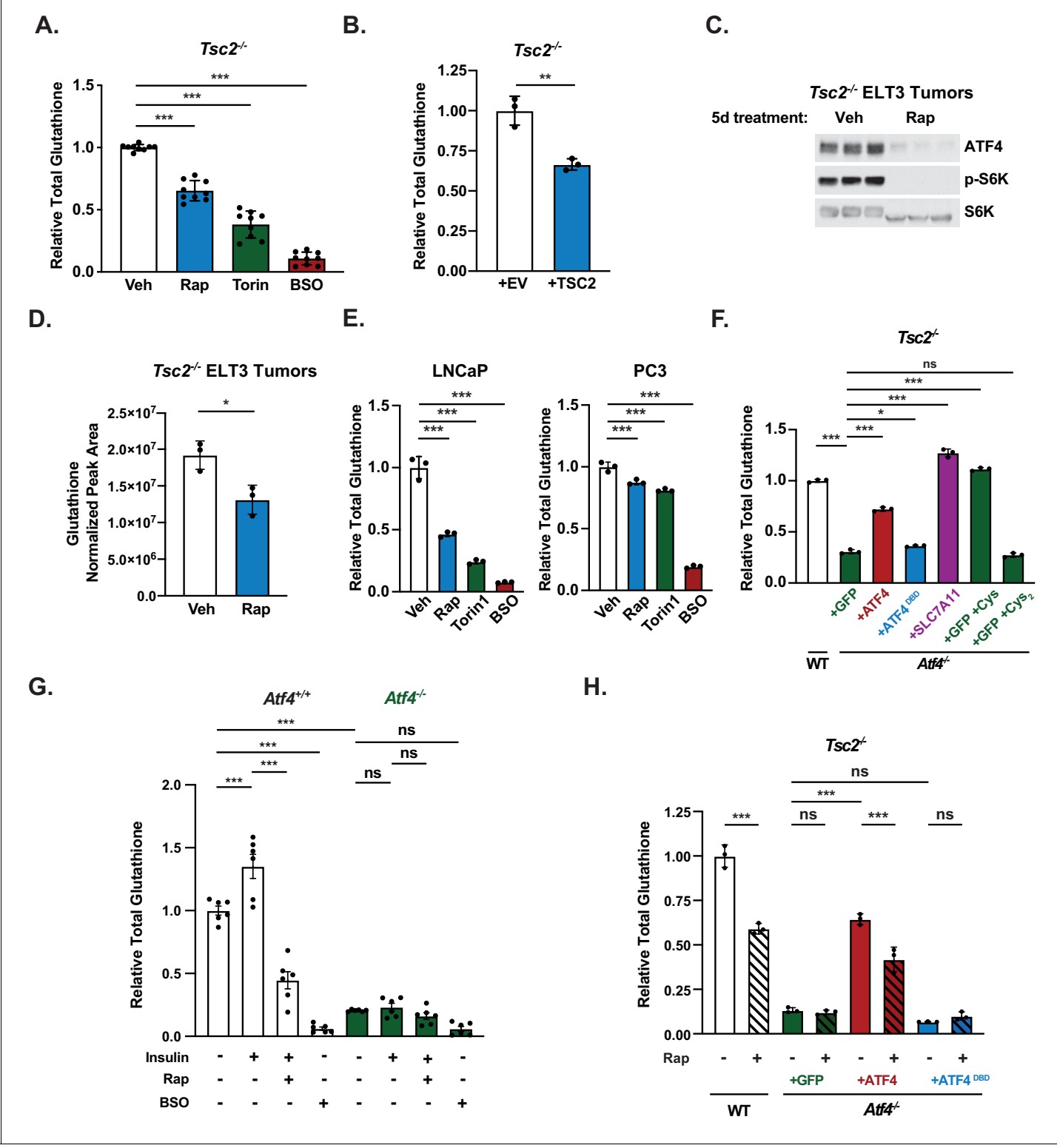

**Figure 7.** Mechanistic target of rapamycin complex 1 (mTORC1) regulates cellular glutathione levels through activating transcription factor 4 (ATF4) and SLC7A11-mediated cystine uptake. (**A**) Total glutathione in serum-deprived *Tsc2*[-/-] mouse embryo fibroblasts (MEFs) treated with rapamycin (20 nM), Torin1 (250 nM), or buthionine-sulfoximine (BSO) (10 μM) for 16 hr is graphed as mean ± SEM relative to vehicle-treated cells from three independent experiments, each with three biological replicates (n = 9). Relative abundance of reduced (GSH) and oxidized (GSSG) glutathione from this experiment is shown in *Figure 7—figure supplement 1A*. (**B**) Relative total glutathione in serum-deprived *Tsc2*[-/-] MEFs with stable reconstitution of a cDNA

*Figure 7 continued on next page*

*Figure 7 continued*

encoding TSC2 or empty vector (EV) control is graphed as mean ± SD from a representative experiment with three biological replicates (n = 3). (**C, D**) Immunoblot (**C**) and relative glutathione levels measured by LC-MS/MS (**D**) from *Tsc2$^{-/-}$* ELT3 xenograft tumors resected from mice treated for 5 days with vehicle or rapamycin (1 mg/kg on days 1, 3, and 5) (n = 3 mice/group). Relative glutathione levels from rapamycin-treated human *TSC2$^{-/-}$* tumor cells are shown in ***Figure 7—figure supplement 1B***. (**E**) Total glutathione in serum-deprived LNCaP (left) and PC3 (right) cells treated with vehicle, rapamycin (20 nM), Torin1 (250 nM), or BSO (50 µM) for 24 hr is graphed as mean ± SD relative to vehicle-treated cells from a representative experiment with three biological replicates (n = 3). (**F**) Total glutathione in serum-deprived *Tsc2$^{-/-}$* MEFs (*Atf4* wild-type [WT]) and *Tsc2$^{-/-}$ Atf4$^{-/-}$* MEFs with stable expression of cDNAs encoding GFP (control), ATF4, ATF4$^{DBD}$, or SLC7A11 grown in Dulbecco's Modified Eagle's Medium and supplemented, where indicated, with cysteine (1 mM, Cys) or cystine (0.5 mM, Cys$_2$) is graphed as mean ± SD relative to WT cells from a representative experiment with three biological replicates (n = 3). Relative glutathione in these cells supplemented with nonessential amino acid with or without Cys, measured by LC-MS/MS, is shown in ***Figure 7—figure supplement 1C***. (**G**) Total glutathione in serum-deprived *Atf4$^{+/+}$* and *Atf4$^{-/-}$* MEFs pretreated 30 min with vehicle or rapamycin (20 nM) prior to insulin stimulation (500 nM, 24 hr) or treated with BSO (10 µM, 24 hr) is graphed as mean ± SEM relative to unstimulated *Atf4$^{+/+}$* cells from two independent experiments, with three biological replicates each (n = 6). (**H**) Total glutathione in serum-deprived *Tsc2$^{-/-}$* MEFs (*Atf4* WT) and *Tsc2$^{-/-}$ Atf4$^{-/-}$* MEFs with stable expression of cDNAs encoding GFP, ATF4 lacking its 5'-UTR, or a DNABD mutant (DBD) of this ATF4 treated with vehicle or rapamycin (20 nM) for 16 hr is graphed as mean ± SD relative to vehicle-treated WT cells from a representative experiment with three biological replicates (n = 3). Effects of mTORC1 signaling and ATF4 on GCLC and GCLM transcript and protein levels are shown in ***Figure 7—figure supplement 1D–G***. (**B, E, F, H**) are representative of at least two independent experiments. *p<0.05, **p<0.01, ***p<0.001, ns = not significant, calculated in (**A, E, F, G, H**) via one-way analysis of variance with Holm–Sidak method for multiple comparisons and in (**B, D**) via unpaired two-tailed *t*-test.

The online version of this article includes the following figure supplement(s) for figure 7:

**Figure supplement 1.** Supplemental data supporting *Figure 7*.

---

leads to a more modest increase in ATF4 protein levels than does the ISR, the selective induction of these genes might be reminiscent of the dose-dependent activation of MYC target genes documented in other studies (*Sabò et al., 2014*; *Schuhmacher and Eick, 2013*; *Walz et al., 2014*). Our bioinformatic analyses and functional data also indicate involvement of the C/EBP family of transcription factors as heterodimerization partners of ATF4 for the regulation of these gene targets shared between mTORC1 signaling and the ISR.

Consistent with the specific ATF4 target genes induced by mTORC1 signaling, including those involved in amino acid acquisition and tRNA charging, we found that ATF4 activation contributes to both canonical (e.g., protein synthesis) and new (e.g., glutathione synthesis) functions of mTORC1. As mTORC1 stimulates protein synthesis through multiple downstream targets (*Holz et al., 2005*; *Jefferies et al., 1994*; *Ma and Blenis, 2009*; *Raught et al., 2004*; *Thoreen et al., 2012*), it was not surprising to find that ATF4 was necessary but not sufficient for the increased rate of protein synthesis accompanying mTORC1 activation. We also demonstrate that mTORC1 signaling regulates the abundance of total cellular glutathione, both reduced and oxidized, at least in part through the ATF4-dependent induction of the cystine transporter SLC7A11, a major source of the cysteine that is limiting for glutathione synthesis. Importantly, this mTORC1-ATF4-mediated transcriptional upregulation of SLC7A11 leading to increased cystine uptake would temporally follow the inhibition of SLC7A11 recently found to be mediated through mTORC2 and Akt-dependent transient phosphorylation of the transporter (*Gu et al., 2017*; *Lien et al., 2017*). Our findings are consistent with a recent study indicating rapamycin-sensitive expression of xCT in TSC models, which the authors attribute to the OCT1 transcription factor (*Li et al., 2019*). However, our study indicates that the mTORC1-mediated activation of ATF4 is both necessary and sufficient for this regulation. The transcription factor NRF2 (also known as NFE2L2) is activated by oxidative stress and is a master regulator of the enzymes required for glutathione synthesis, as well as SLC7A11 to increase cystine uptake (*Habib et al., 2015*; *Sasaki et al., 2002*; *Ye et al., 2014*). While NRF2 depletion has been described to decrease the viability of cells with TSC gene loss (*Zarei et al., 2019*), we have no evidence from this or previous studies that mTORC1 signaling influences the levels or activity of NRF2 (*Zhang et al., 2014*). mTORC1 serves to couple growth signals to the coordinated control of anabolic processes, including the biosynthesis of protein, lipids, and nucleotides, as well as metabolic pathways that support this anabolic state (*Valvezan and Manning, 2019*). This metabolic program is orchestrated to provide biosynthetic precursors and directly promote the synthesis of macromolecules while also maintaining cellular homeostasis and preventing nutrient or metabolic stress. For example, mTORC1 signaling promotes metabolic flux through the NADPH-producing oxidative branch of the pentose phosphate pathway, thereby providing the reducing power essential to

support an mTORC1-stimulated increase in de novo lipid synthesis (*Düvel et al., 2010*). Importantly, NADPH is also essential to reduce cystine, taken up through SLC7A11, into two molecules of cysteine for use in glutathione synthesis, in addition to being required to regenerate reduced glutathione following its oxidation. Supporting this logic of a coordinated metabolic program downstream of mTORC1, pro-growth signaling through mTORC1 likely promotes glutathione synthesis to help buffer against the oxidative stress that accompanies anabolic metabolism and increased rates of protein synthesis (*Han et al., 2013*; *Harding et al., 2003*; *Kong and Chandel, 2018*).

Our findings further support the addition of ATF4 to SREBP and HIF1, as nutrient- and stress-sensing transcription factors that are independently co-opted by mTORC1 signaling to drive the expression of metabolic enzymes and nutrient transporters. Unlike adaptive signals stemming from the depletion of individual nutrients, such as amino acids, sterols, or oxygen, which generally attenuate mTORC1 signaling as part of the adaptive response, pro-growth signals that activate mTORC1 can stimulate these transcription factors in concert to support a broader anabolic program. It will be important in future studies to understand the dual regulation of these transcription factors by both pro-growth and adaptive mechanisms as it relates to settings of physiological (fasting and feeding) and pathological (tumor development) nutrient fluctuations.

# Materials and methods

**Key resources table**

| Reagent type (species) or resource | Designation | Source or reference | Identifiers | Additional information |
|---|---|---|---|---|
| Biological sample (*Mus musculus*) | ELT3 tumor samples | PMID:29056426 | | *Valvezan et al., 2017* |
| Cell line (*M. musculus*) | WT and *Tsc2*$^{-/-}$ MEFs | PMID:14561707 | | David Kwiatkowski |
| Cell line (*M. musculus*) | *EIF2a*$^{A/A}$ MEFs | PMID:11430820 | | Randal Kaufman |
| Cell line (*M. musculus*) | *Rictor*$^{+/+}$ and *Rictor*$^{-/-}$ MEFs | PMID:17141160 | | D.A. Guertin and D.M. Sabatini |
| Cell line (*Homo sapiens*) | PC3 | ATCC | CRL-1435 RRID:CVCL_0035 | |
| Cell line (*H. sapiens*) | LNCaP | ATCC | CRL-1740 RRID:CVCL_1379 | |
| Cell line (*M. musculus*) | *Tsc2*$^{+/+}$ *Atf4*$^{-/-}$ MEFs | This paper | | CRISPR-Cas9n generated – see Materials and methods |
| Cell line (*M. musculus*) | *Tsc2*$^{-/-}$ *Atf4*$^{-/-}$ MEFs | This paper | | CRISPR-Cas9n generated – see Materials and methods |
| Transfected construct (*Aequorea victoria*) | pTRIPZ-EGFP | This paper | | eGFP cDNA- expressing control plasmid – see Materials and methods |
| Transfected construct (*M. musculus*) | pTRIPZ-ATF4 | This paper | | Rapamycin-resistant ATF4 cDNA- expressing plasmid – see Materials and methods |
| Transfected construct (*M. musculus*) | pTRIPZ-ATF4$^{DBD}$ | This paper | | ATF4 DNA-binding domain mutant cDNA-expressing plasmid – see Materials and methods |
| Transfected construct (*H. sapiens*) | pTRIPZ-SLC7A11 | This paper | | SLC7A11 cDNA- expressing plasmid – see Materials and methods |
| Transfected construct (human, mouse) | Non-targeting pool for siRNA experiments | GE Life Sciences/ Dharmacon | D-001810-10-50 | |
| Transfected construct (mouse) | siMyc | GE Life Sciences/ Dharmacon | L-040813-00-0010 | |

*Continued on next page*

*Continued*

| Reagent type (species) or resource | Designation | Source or reference | Identifiers | Additional information |
|---|---|---|---|---|
| Transfected construct (mouse) | siAtf4 | GE Life Sciences/ Dharmacon | L-042737-01-0020 | |
| Transfected construct (mouse) | siRheb | GE Life Sciences/ Dharmacon | L-057044-00-0020 | |
| Transfected construct (mouse) | siRhebL1 | GE Life Sciences/ Dharmacon | L-056074-01-0020 | |
| Transfected construct (mouse) | siTsc2 | GE Life Sciences/ Dharmacon | L-047050-00-0020 | |
| Transfected construct (mouse) | siC/ebpα | GE Life Sciences/ Dharmacon | L-040561-00-0005 | |
| Transfected construct (mouse) | siC/ebpβ | GE Life Sciences/ Dharmacon | L-043110-00-0005 | |
| Transfected construct (mouse) | siC/ebpδ | GE Life Sciences/ Dharmacon | L-060294-01-0005 | |
| Transfected construct (mouse) | siC/ebpγ | GE Life Sciences/ Dharmacon | L-065627-00-0005 | |
| Transfected construct (human) | siATF4 | GE Life Sciences/ Dharmacon | L-005125-00-0020 | |
| Sequenced-based reagent | qPCR primers | IDT | | See table in Materials and methods |
| Recombinant DNA reagent | ATF4 (cDNA amplified from plasmid) | Addgene | RRID:Addgene_21845 | |
| Recombinant DNA reagent | Pspax2 (plasmid) | Addgene | RRID:Addgene_12260 | |
| Recombinant DNA reagent | Pmd2.G (plasmid) | Addgene | RRID:Addgene_12259 | |
| Recombinant DNA reagent | pSpCas9n(BB)−2A-GFP (PX461) (plasmid) | Addgene | RRID:Addgene_48140 | |
| Recombinant DNA reagent | pTRIPZ (plasmid) | PMID:27088857 | | Alex Toker (Beth Israel Deaconess Medical Center) |
| Recombinant DNA reagent | GFP (cDNA amplified from plasmid) | Addgene | RRID:Addgene_19319 | |
| Recombinant DNA reagent | SLC7A11 (cDNA amplified from plasmid) | PMID:29259101 | | Alex Toker (Beth Israel Deaconess Medical Center) |
| Recombinant DNA reagent | pBabe hygro IRES-TSC2 | PMID:15150095 | | David Kwiatkowski (Brigham and Women's Hospital) |
| Sequenced-based reagent | CRISPR-Cas9n guides for KO of ATF4 | IDT | | CACCGGAGG TGGAGGGGCTATGCT; AAACAGCATAGCCCC TCCACCTCC; CACCGA-CAATCTGCCTTC TCCAGG; AAACCC TGGAGAAGGCAGATTG TC |
| Sequenced-based reagent | Sequencing primers for Atf4-/- cell lines | IDT | | TCGATGCTCTGTTTCGAA TG; CTTCTTCCCCC TTGCCTTAC |

*Continued on next page*

Continued

| Reagent type (species) or resource | Designation | Source or reference | Identifiers | Additional information |
|---|---|---|---|---|
| Sequenced-based reagent | Primers for site-directed mutagenesis | IDT | | GCCTCCTGC TCAGCCGCCGCCGCC TCGAGGTACCCAGTGGC TGCTGTCTTGTTTTGC TCCATCT; AGATGGAG-CAAAACAAGACAG-CAGCCACTGGGTACC TCGAGGCGGCGGCGGC TGAGCAGGAGGC |
| Commercial assay or kit | KOD Xtreme Hot Start DNA Polymerase | Sigma-Aldrich | 71975 | |
| Antibody | (P)-S6K1 T389 rabbit monoclonal | Cell Signaling Technologies (CST) | Cat #: 9234 RRID:AB_2269803 | 1:1000, 10 μL |
| Antibody | ATF4 rabbit monoclonal | Cell Signaling Technologies (CST) | Cat #: 11815 RRID:AB_2616025 | 1:1000, 10 μL |
| Antibody | eIF2α rabbit polyclonal | Cell Signaling Technologies | Cat #: 9722 RRID:AB_2230924 | 1:1000, 10 μL |
| Antibody | P-eIF2α S51 rabbit polyclonal | Cell Signaling Technologies | Cat #: 9721 RRID:AB_330951 | 1:1000, 10 μL |
| Antibody | S6K1 rabbit monoclonal | Cell Signaling Technologies | Cat #: 2708 RRID:AB_390722 | 1:1000, 10 μL |
| Antibody | CD98 rabbit monoclonal | Cell Signaling Technologies | Cat #: 13180 RRID:AB_2687475 | 1:1000, 10 μL |
| Antibody | PSAT1 rabbit polyclonal | Protein Tech | Cat #: 20180-1-AP RRID:AB_10665948 | 1:1000, 10 μL |
| Antibody | MTHFD2 rabbit polyclonal | Protein Tech | Cat #: 12270–1-AP RRID:AB_2147525 | 1:1000, 10 μL |
| Antibody | AARS rabbit polyclonal | Bethyl Antibodies | A303-475A-M | 1:1000, 10 μL |
| Antibody | GARS rabbit polyclonal | Bethyl Antibodies | A304-746A-M | 1:1000, 10 μL |
| Antibody | LARS rabbit polyclonal | Bethyl Antibodies | A304-316A-M | 1:1000, 10 μL |
| Antibody | XPOT rabbit polyclonal | Aviva Biotechnologies | Cat #: ARP40711_P050 RRID:AB_2048757 | 1:1000, 10 μL |
| Antibody | TSC2 rabbit monoclonal | Cell Signaling Technologies | Cat #: 4308 RRID:AB_10547134 | 1:1000, 10 μL |
| Antibody | SLC7A11 rabbit monoclonal | Abcam | Cat #: ab175186 RRID:AB_2722749 | 1:1000, 10 μL For immunoblots of WT and $Tsc2^{-/-}$ MEFs |
| Antibody | SLC7A11 rabbit monoclonal | Cell Signaling Technologies | Cat #: 12691 RRID:AB_2687474 | 1:1000, 10 μL For immunoblots of PC3 and LNCaP cell lines |
| Antibody | GCLC rabbit monoclonal | Abcam | ab190685 | 1:1000, 10 μL |
| Antibody | GCLM rabbit monoclonal | Abcam | Cat#: ab126704 RRID:AB_11127439 | 1:1000, 10 μL |
| Antibody | RICTOR rabbit monoclonal | Cell Signaling Technologies | Cat #: 9476 RRID:AB_10612959 | 1:1000, 10 μL |
| Antibody | RHEB rabbit monoclonal | Cell Signaling Technologies | Cat #: 13879 RRID:AB_2721022 | 1:1000, 10 μL |
| Antibody | 4EBP1 rabbit monoclonal | Cell Signaling Technologies | Cat #: 9644 RRID:AB_2097841 | 1:1000, 10 μL |
| Antibody | c-MYC rabbit polyclonal | Cell Signaling Technologies | Cat #: 9402 RRID:AB_2151827 | 1:1000, 10 μL |
| Antibody | β-Actin mouse monoclonal | Sigma | Cat #: A5316 RRID:AB_476743 | 1:5000, 2 μL |

*Continued*

| Reagent type (species) or resource | Designation | Source or reference | Identifiers | Additional information |
|---|---|---|---|---|
| Antibody | HRP-conjugated anti-rabbit rabbit polyclonal | CST | Cat #: 7074 RRID:AB_2099233 | 1:5000, 2 µL |
| Antibody | HRP-conjugated anti-mouse mouse polyclonal | CST | Cat #: 7076 RRID:AB_330924 | 1:5000, 2 µL |
| Antibody | IRDye 800CW Donkey anti-Rabbit IgG rabbit polyclonal | LI-COR | Cat #: 925–32213 RRID:AB_2715510 | 1:5000, 2 µL |
| Antibody | IRDye 800CW Donkey anti-Mouse IgG mouse polyclonal | LI-COR | Cat #: 926-32212 RRID:AB_621847 | 1:5000, 2 µL |
| Chemical compound, drug | Doxycycline hydrochloride | Sigma | D3447 | |
| Chemical compound, drug | Rapamycin | LC Laboratories | R5000 | |
| Chemical compound, drug | Insulin | Alpha Diagnostic, | INSL 16 N-5 | |
| Chemical compound, drug | Tunicamycin | Sigma-Aldrich | T7765 | |
| Chemical compound, drug | Torin1 | Tocris | 4247 | |
| Chemical compound, drug | Erastin | Selleckchem | S7242 | |
| Chemical compound, drug | Buthionine-sulfoximine (BSO) | Sigma | B2515 | |
| Chemical compound, drug | Hygromycin B | Thermo Fisher Scientific | 10687010 | |
| Chemical compound, drug | Puromycin | Sigma | P8833 | |
| Chemical compound, drug | Staurosporine | Tocris | 1285 | |
| Chemical compound, drug | Cysteine | Sigma | C7477 | |
| Chemical compound, drug | Cystine | Sigma | 57579 | |
| Chemical compound, drug | 2-Mercaptoethanol | EMD Millipore | 444203 | |
| Chemical compound, drug | MEM Nonessential amino acids solution | Thermo Fisher Scientific | 11140050 | |
| Chemical compound, drug | $^{35}$S-methionine | PerkinElmer | NEG009L005MC | |
| Chemical compound, drug | L-[1, 2, 1′, 2′-$^{14}$C]-Cystine | PerkinElmer | NEC854010UC | |
| Commercial assay or kit | FITC Annexin V Apoptosis Detection Kit I | BD | 556547 | |
| Commercial assay or kit | GSH/GSSG-Glo Assay | Promega | V6611 | |

## Cell culture

MEFs and PC3 cells were maintained in DMEM (Corning/Cellgro, 10-017-CV) with 10% fetal bovine serum (FBS, Corning/Gibco). LNCaP cells were maintained in RPMI-1640 (Corning/Cellgro 10-040-CV) with 10% FBS. *Tsc2*$^{-/-}$ (*p53*$^{-/-}$) MEFs and littermate-derived wild-type counterparts were provided by David Kwiatkowski (Brigham and Women's Hospital, Boston, MA). *eIF2α*$^{S/S}$ (WT) and *eIF2α*$^{A/A}$ (S51A knock-in mutant) MEFs were provided by Randal Kaufman (Sanford-Burnham-Prebys Medical Discovery Institute, La Jolla, CA) and were not used above passage 3 (after received). *Rictor*$^{+/+}$ and *Rictor*$^{-/-}$ MEFs were provided by D.A. Guertin and D.M. Sabatini (Whitehead Institute, Massachusetts Institute of Technology, Cambridge, MA). Cancer cell lines were obtained from ATCC. *Atf4*$^{-/-}$ MEF lines generated in this study were maintained in DMEM with 10% FBS, supplemented with 55 µM 2-mercaptoethanol (Thermo, 21985023), and 1X MEM NEAA mix (NEAA, final concentrations: 100 µM each of alanine, aspartate, asparagine, glutamate, glycine, proline, and serine; Thermo 11140050). In experiments with supplementation of excess cysteine, cells were plated in DMEM with 10% FBS, 1 mM cysteine, and, where indicated, 1X MEM NEAA mix.

## siRNA knockdowns

Cells were transfected with 20 nM of the indicated siRNAs using Opti-MEM (Thermo, 31985062) and RNAimax (Thermo, 13778150) according to the manufacturer's protocol. siRNAs were from GE Life

Sciences/Dharmacon: non-targeting pool (D-001810-10-50), *Myc* (L-040813-00-0010), *Atf4* (mouse, L-042737-01-0020), *Rheb* (L-057044-00-0020), *RhebL1* (L-056074-01-0020), *Tsc2* (L-047050-00-0020), C/ebpα (L-040561-00-0005), C/ebpβ (L-043110-00-0005), C/ebpδ (L-060294-01-0005), C/ebpγ (L-065627-00-0005), and *ATF4* (human, L-005125-00-0020). Forty-eight hours post transfection, cells were treated as indicated prior to lysis for immunoblotting, RNA extraction, or protein synthesis assays. For C/EBP isoform knockdown experiments, transfection of siRNAs was performed a second time, 24 hr after the first transfection. For protein synthesis assays involving siRNAs, transfection was also performed a second time, 24 hr after the first transfection, which was necessary to achieve sufficient knockdown of Rheb to reduce mTORC1 signaling.

## Generation and validation of *Atf4*⁻/⁻ and reconstituted cell lines

*Tsc2*⁺/⁺ (WT) and *Tsc2*⁻/⁻ MEFs lacking Atf4 were generated by CRISPR-Cas9-mediated deletion using pSpCas9n(BB)−2A-GFP (PX461) vector (Addgene, 48140) according to the previously described protocol (*Ran et al., 2013*). The paired nickase guides were designed using E-CRISP (*Heigwer et al., 2014*) and targeted the sequences AGCATAGCCCCTCCACCTCC and GACAATC TGCCTTCTCCAGG in exon 2 of ATF4. Forty-eight hours post transfection, single GFP-positive cells were sorted into 96-well plates. Cells were cultured in DMEM with 10% FBS supplemented with 1X MEM NEAA and 55 µM 2-mercaptoethanol. Single cell clones were grown for immunoblot analysis, and those showing loss of ATF4 protein were selected for sequence analysis involving the isolation of genomic DNA (Qiagen, 69504), PCR amplification using KOD Xtreme Hot Start DNA Polymerase (Millipore, 71975), and the primers TCGATGCTCTGTTTCGAATG and CTTCTTCCCCCTTGCCTTAC flanking the targeted deletion site, with sequencing on an ABI3730xl DNA analyzer at the DNA Resource Core of Dana-Farber/Harvard Cancer Center (funded in part by NCI Cancer Center support grant 2P30CA006516-48). The mutations were identified using CRISP-ID software (*Dehairs et al., 2016*). For the final clones selected, the mutations generated in the WT MEFs include an out-of-frame 17-bp deletion starting at the codon encoding T237 and a large out-of-frame 73-bp deletion starting after the codon encoding G219, both resulting in premature STOP codons. The mutations generated in the *Tsc2*⁻/⁻ MEFs include an out-of-frame 245-bp deletion starting at the codon encoding G190, resulting in a premature STOP codon after the D191 codon, and an in-frame deletion removing the sequences encoded between E210 and E284.

For generation of *Atf4* expression vectors, the murine *Atf4* cDNA was amplified from the plasmid 21845 from Addgene (*Harding et al., 2000*). Restriction enzyme cloning with AgeI and ClaI was used to insert the *Atf4*-coding sequence (lacking the 5′ and 3′-UTR) into the pTRIPZ plasmid for doxycycline-inducible expression. The ATF4ᴰᴮᴰ mutant, in which amino acids 292–298 of the DNA-binding domain are changed from RYRQKKR to GYLEAAA (*He et al., 2001*; *Lange et al., 2008*), was generated by DpnI-mediated site-directed mutagenesis using KOD Xtreme Hot Start DNA Polymerase. The dox-inducible pTRIPZ and SLC7A11 plasmids were a gift from Alex Toker (Beth Israel Deaconess Medical Center, Boston, MA). cDNA expression was induced with 1 µg/mL of doxycycline (Sigma-Aldrich, D3447) for 12–24 hr before assays were conducted. GFP was inserted into pTRIPZ to produce the control vector. Lentivirus was generated in HEK293T cells transfected with pMD2.G and psPAX2 (Addgene, 12259 and 12260) and the given pTRIPZ constructs. Forty-eight hours post transfection, the virus-containing medium was used to infect the *Atf4* knockout cells, which were selected with 8 µg/mL puromycin. TSC2 addback cell lines were generated by retroviral infection following transfection of PT67 cells with pBabe hygro IRES-EV or pBabe hygro IRES-TSC2. Cells were selected with 400 µg/mL hygromycin B (Thermo, 10687010).

## RNA-sequencing

Wild-type and *Atf4*⁻/⁻ MEFs were grown to 70% confluence in 6 cm plates and were serum-starved in the presence of 2-mercaptoethanol and 1X MEM NEAA mixture and treated with vehicle (DMSO) or 20 nM rapamycin (LC Laboratories, R5000) for 30 min prior to stimulation with vehicle (water) or 500 nM insulin (Alpha Diagnostic, INSL 16 N-5) for 16 hr or treated with vehicle (DMSO) or 2 µg/mL tunicamycin (Sigma-Aldrich, T7765) for 4, 8, or 16 hr. RNA was harvested with TRIzol according to the manufacturer's protocol (Thermo, 15596018). All samples passed RNA quality control measured by NanoDrop 1000 Spectrophotometer (NanoDrop Technologies) and 2100 Bioanalyzer (Agilent Technologies). cDNA libraries were generated to produce 150-bp paired-end reads on an Illumina

NovaSeq with read depth of 20 million paired reads per sample. Reads were aligned and annotated to the Ensembl *Mus musculus* GRCm38.p6 genome assembly using the align and featureCounts functions from the Rsubread package (2.0.1) in R (3.6.3) (*Liao et al., 2019*). Differential gene expression analysis was performed using the voom and eBayes functions from the EgdeR (3.28.1) and Limma (3.42.2) packages, respectively (*Ritchie et al., 2015*; *Robinson et al., 2010*). Transcripts found to be significantly induced by tunicamycin were further limited to those with a greater than 1.2-fold increase. The enrichKEGG function from the clusterProfiler package (3.14.3) was used to perform KEGG pathway over-representation tests (*Yu et al., 2012*). Gene set enrichment analysis was evaluated using GSEA software from the Broad Institute (*Subramanian et al., 2005*). Computations were run on the FASRC Cannon cluster, supported by the Faculty of Arts and Sciences Division of Science Research Computing Group at Harvard University. Pseudogenes and unannotated genes were excluded from *Figure 1C* and *Figure 1—source data 1*. The complete RNA-seq data can be found at GEO under the accession number GSE158605.

## CiiiDER analysis

CiiiDER software was downloaded from CiiiDER.org with the *M. musculus* GRCm38.94 genome files. Searches were run against the JASPAR transcription factor-binding profile database. Searches were run on promoter regions spanning +1500 to −500 bp from the predicted transcriptional start site using a site identification deficit threshold of 0.1. The background gene list (ISR only) comprised the 200 ATF4-dependent genes most significantly increased in expression upon tunicamycin treatment that were not in the list of 61 genes shared in their regulation by mTORC1 signaling and the ISR. Results of this analysis are included in *Figure 2—source data 1*.

## Cistrome analysis

Each of the 61 shared mTORC1 and ISR genes was analyzed using the CistromeDB Toolkit (http://dbtoolkit.cistrome.org/) of existing genome-wide ChIP-seq data. A half-decay distance of 1 kb to the transcription start site was used. The top 20 transcription factors or chromatin regulators found in ChIP-seq experiments to bind to each gene were compiled, and the number of genes each factor bound to within the list of 61 was determined, with the top 9 regulators graphed, excluding the general factors EP300 and POL2RA.

## Immunoblotting

Cells were lysed in ice-cold Triton lysis buffer (40 mM HEPES pH 7.4, 120 mM NaCl, 1 mM EDTA, 1% Triton X-100, 10 mM sodium pyrophosphate, 10 mM glycerol 2-phosphate, 50 mM NaF, 0.5 mM sodium orthovanadate, 1 μM Microcystin-LR, and Sigma protease inhibitor cocktail P8340). For immunoblots on SLC7A11, cells were lysed in 1% SDS lysis buffer (10 mM Tris pH 7.4, 1 mM EDTA, 10 mM sodium pyrophosphate, 10 mM glycerol 2-phosphate, 50 mM NaF, 0.5 mM sodium orthovanadate, and Sigma protease inhibitor cocktail P8340). Samples were centrifuged at 20,000 × *g* for 10 min at 4°C, and protein concentration in the supernatant was determined by Bradford assay (Bio-Rad, 5000202) and normalized across samples. Proteins were separated by SDS-PAGE, transferred to nitrocellulose membranes, and immunoblotted with indicated antibody. Primary antibodies used were MTHFD2 (Protein Tech, 12270-1-AP), PSAT1 (Protein Tech 20180-1-AP), phospho (P)-S6K1 T389 (Cell Signaling Technologies [CST], 9234), ATF4 (CST, 11815), eIF2α (CST, 9722), P-eIF2α S51 (CST, 9721), S6K1 (CST, 2708), CD98 (CST, 13180), AARS (Bethyl Antibodies, A303-475A-M), GARS (Bethyl Antibodies, A304-746A-M), LARS (Bethyl Antibodies, A304-316A-M), XPOT (Aviva Biotechnologies, ARP40711_P050), TSC2 (CST, 4308), SLC7A11 (mouse, Abcam, ab175186), SLC7A11 (human, CST, 12691), GCLC (Abcam, ab190685), GCLM (Abcam, ab126704), RICTOR (CST, 9476), RHEB (CST, 13879), 4EBP1 (CST 9644), c-MYC (CST, 9402), and β-actin (Sigma-Aldrich, A5316); secondary antibodies used were IRDye 800CW Donkey anti-Mouse IgG (H + L) (LI-COR, 926-32212) and Donkey anti-Rabbit IgG (H + L) (LI-COR, 925–32213), and HRP-conjugated anti-mouse and anti-rabbit secondary antibodies from (CST, 7074 and 7076). Immunoblots of MTHFD2, PSAT1, AARS, GARS, LARS, and XPOT were imaged using Odyssey CLx Imaging System (LI-COR Biosciences). β-Actin was developed with Odyssey CLx Imaging System or enhanced chemiluminescence assay (ECL). All remaining immunoblots were developed using ECL. Immunoblots were quantified using the Odyssey CLx Imaging System and were normalized to β-actin. The ATF4 immunoblot

corresponding to *Figure 1B* (*Figure 1—figure supplement 1A*) and the SLC7A11 immunoblots in *Figure 6D, E* (*Figure 6—figure supplement 1B, C*) was quantified using ImageJ and normalized to β-actin.

## NanoString analysis

RNA was harvested using TRIzol from cells at 70% confluence. All samples passed RNA quality control measured by NanoDrop 1000 Spectrophotometer. Parallel plates were lysed for immunoblots. The isolated RNA was analyzed using a custom NanoString probe library according to the manufacturer's instructions (NanoString Technologies). Briefly, sample RNA was hybridized to RNA-probes at 65°C for 16 hr, excess probe was washed away, and an nCounter SPRINT was used to quantify specific mRNA molecules present in each sample. Direct mRNA counts were normalized to internal control genes, and mRNA expression was analyzed using nSOLVER software. Heatmaps were generated with Morpheus software from the Broad Institute (https://software.broadinstitute.org/morpheus). Transcripts with 100 counts or fewer, a value based off of negative control samples, were not included in our analyses. Results of this analysis are included in *Figure 3—source data 1*.

## qPCR

For gene expression analysis, RNA was isolated with TRIzol according to the manufacturer's protocol. All samples passed RNA quality control measured by NanoDrop 1000 Spectrophotometer. cDNA was generated with Superscript III First Strand Synthesis System for RT-PCR (Thermo, 18080051). Quantitative RT-PCR was performed with a CFX Connect Realtime PCR Detection System (Bio-Rad) using iTaq Universal SYBR Green Supermix (Bio-Rad, 1725125). Samples were quantified by the ΔΔCT method, normalized to *β-actin* (mouse samples) or *RPLP0* (human samples) to quantify relative mRNA expression levels. qPCR primers:

| | | |
|---|---|---|
| *Atf4* | GATGGGTTCTCCAGCGACAAG | CCGGAAAAGGCATCCTCCTTC |
| *Psat1* | GCTGTCGCCTTAGCACCA | TGGATCTCCAACAATACCGAGTG |
| *Mthfd2* | TCCTTGTTGTCTGCGTTGGC | CTTCATTTCGCACTGCCGCC |
| *Slc7a5* | GGACAAGGTGATGCGTCCAA | GCCAACACAATGTTCCCCAC |
| *Aars* | TTGCTATTCCCTCGGAGCAC | CTCCTCGGGAACCTTAGCTC |
| *Gars* | GGCAGAGGTCTCTGAGCTG | GCACGATGGTCATAAGCTGC |
| *Cars* | GAGCAGGCTGCCGACTACA | TATAGCTACGCGTGCTGAGG |
| *Nars* | GAGCCGGCCTGTGTAAAGAT | GACCCAGCCAAACACCTTCA |
| *Iars* | TATTGCATCACCTCCAGACGC | TGAACCATTCTGTTGCTGGGA |
| *Gclc* | TGTACTCCACCCTCGTCACCC | CTGCTGTCCCAAGGCTCG |
| *Gclm* | TGGGCACAGGTAAAACCCAA | CTGGGCTTCAATGTCAGGGA |
| *Slc7a11* | ATCTCCCCCAAGGGCATACT | GCATAGGACAGGGCTCCAAA |
| *Slc3a2* | TGATGAATGCACCCTTGTACTTG | GCTCCCCAGTGAAAGTGGA |
| *Slc1a5* | GTAAAATACCGCAATCCTGTATCC | CGATAGCGAAGACCACCAGG |
| *Xpot* | GCTTCAGGCTCAGATGCAGA | AAAGCAAGGCGAACACTTGG |
| *c-Myc* | AGAGCTCCTCGAGCTGTTTG | TTCTCTTCCTCGTCGCAGAT |
| *C/ebpα* | CAAGAACAGCAACGAGTACCG | GTCACTGGTCAACTCCAGCAC |
| *C/ebpβ* | CGCCTTATAAACCTCCCGCT | TGGCCACTTCCATGGGTCTA |
| *C/ebpδ* | CGACTTCAGCGCCTACATTGA | CTAGCGACAGACCCCACAC |
| *C/ebpγ* | TCGGATCACATTGCTCTGATTTC | TGTGCCTGAGTATGAATGACACT |
| *Actin* | CACTGTCGAGTCGCGTCC | TCATCCATGGCGAACTGGTG |
| *PSAT1* | AAAAACAATGGAGGTGCCGC | GGCTCCACTGGACAAACGTA |
| *ASNS* | TGGCTGCCTTTTATCAGGGG | TCTGCCACCTTTCTAGCAGC |
| *MTHFD2* | GGCAGTTCGAAATGAAGCTGTT | GCCAACCAGGATCACACTCA |

*Continued on next page*

| SLC7A5 | GACTACGCCTACATGCTGGA | AGCAGCAGCACGCAGAG |
|---|---|---|
| AARS | CCATTCAGAAGGGCACAGGT | TATCCACGCCCTGTGTTGTC |
| GARS | GCCAGCAGGGAGATCTTGTG | CCAGCTCCTTTGCTTCCAGA |
| XPOT | GACGCAGAGCGACTAGAGG | TAAACATCTTCCCTATCACTCCATC |
| SLC7A11 | AAGGTGCCACTGTTCATCCC | ATGTTCTGGTTATTTTCTCCGACA |
| RPLP0 | CCTCGTGGAAGTGACATCGT | ATCTGCTTGGAGCCCACATT |

## Protein synthesis assay

Cells were cultured as indicated, washed twice with PBS, and changed to methionine/cystine/glutamine-free DMEM (Thermo, 21013024) supplemented with 0.5 mM L-cystine (Sigma, 57579) and L-glutamine (4 mM) with 50 µCi/mL $^{35}$S-methionine (PerkinElmer, NEG009L005MC) for 20 min. Cells were washed twice in ice-cold PBS and lysed in ice-cold Triton lysis buffer. Total protein concentrations were normalized following a Bradford assay, and normalized samples were separated by SDS-PAGE and transferred to nitrocellulose. $^{35}$S-methionine incorporation into protein was analyzed by autoradiography, and relative rates of protein synthesis were quantified using ImageJ Software (NIH) to quantify radiolabeled protein per lane for each sample. For isogenic $Atf4^{-/-}$ cell lines, exogenous cDNA expression was induced for 16 hr with 150 ng/mL doxycycline, and cells were treated as indicated in the presence of doxycycline (150 ng/mL) and 1X MEM NEAA mixture plus 1 mM cysteine (Sigma, C7477). Protein synthesis was assayed with 20 min labeling in the absence of NEAA and cysteine to avoid competition of $^{35}$S-methionine uptake with the supplemented amino acids.

## Methionine and cystine uptake

For methionine uptake assays, cells were cultured and labeled as described above for the protein synthesis assay, were washed three times in cold PBS, and lysed in Triton lysis buffer. For cystine uptake assays, cells were treated the same but labeled for the final 10 min with medium containing 0.1 µCi L-[1, 2, 1', 2'-$^{14}$C]-Cystine (PerkinElmer, NEC854010UC) and washed three times in ice-cold PBS containing cold cystine (1 mM), prior to lysis in Triton lysis buffer. Whole-cell radiolabel incorporation was quantified with a Beckman LS6500 scintillation counter. Cells from identically treated parallel plates were counted using a Beckman Z1-Coulter Counter to normalize uptake measurements to cell number.

## Proliferation assay

To quantify cell proliferation, cells were plated in DMEM in 6-well plates in triplicate in the presence of 2-mercaptoethanol, 1X MEM NEAA mixture, 10% FBS, and doxycycline (1 µg/mL). Twenty-four hours after plating, cells were washed twice with PBS and media was changed to DMEM with 10% dialyzed FBS, doxycycline (1 µg/mL), and the amino acid supplements indicated for each experiment, with media refreshed daily. Starting on day 0, viable cells from triplicate wells corresponding to each condition or cell line were counted using a hemocytometer, excluding dead cells detected by trypan blue stain (Sigma-Aldrich, T8154).

## Analysis of cell death

Cells were plated in DMEM in 6-well plates in triplicate in the presence of 2-mercaptoethanol, 1X MEM NEAA mixture, 10% FBS, and doxycycline (1 µg/mL). Twenty-four hours after plating, cells were washed twice with PBS and media was changed to DMEM with 10% dialyzed FBS and doxycycline (1 µg/mL). Seventy-two hours later, cells were detached with Accumax (Sigma-Aldrich, A7089) and washed twice with cold PBS on ice. Cells were stained with annexin V and propidium iodide (PI) according to the manufacturer's instruction (BD, 556547). Samples were analyzed using an LSRFortessa (BD) flow cytometer, and the fractions of stained cells were quantified using FloJo 10.6, with Staurosporine (4 hr, 5 µM) (Tocris, 1285) used as a positive control for cell death and to help establish gating of the sorted cells.

## Measurements of cellular and tumor glutathione

Cells were plated in 96-well plates at 5000 cells/well. Total glutathione, GSH, and GSSG levels were measured using the GSH/GSSG-Glo Assay (Promega, V6611) according to the manufacturer's protocol. Total glutathione levels were normalized to cell number determined from parallel plates. BSO (Sigma, B2515) was used as a positive control to inhibit glutathione synthesis.

For measurements via LC-MS/MS, metabolites were extracted from cells on dry ice using 80% methanol, and extracts were dried under nitrogen gas for metabolite profiling via selected reaction monitoring with polarity switching using a 5500 QTRAP mass spectrometer. Data were analyzed using MultiQuant 2.1.1 software (AB/SCIEX) to calculate the Q3 peak area. Normalized peak area of glutathione from human $TSC2^{-/-}$ angiomyolipoma (621-101) cells was determined from previously published data (*Tang et al., 2019*). For xenograft tumor studies, experimental details were provided previously (*Valvezan et al., 2017*). Briefly, mice bearing $TSC2^{-/-}$ ELT3 xenograft tumors were treated every other day for 5 days with vehicle or rapamycin (1 mg/kg on days 1, 3, and 5) and tumors were harvested for metabolite extraction, as above, 3 hr after the final treatment.

## Statistics

For RNA-sequencing analysis, Benjamini–Hochberg false discovery rate (FDR)-adjusted p values were determined from empirical Bayes moderated t-statistics using the voom and eBayes functions from the limma package. Comparisons with FDR-adjusted $p<0.05$ were considered significant for the gene groups denoted compared to vehicle-treated controls. For KEGG enrichment, p values were FDR corrected. For CiiiDER transcription factor over-representation analysis, Fisher's exact test p values were used. Transcription factor binding elements with $p<0.01$ and test statistic $>0$ were considered over-represented in genes of interest. Unpaired two-tailed *t*-tests were used for NanoString analyses to calculate p values for rank ordering. All remaining statistical analyses were performed with Prism 8 software (GraphPad Software, La Jolla, CA). Statistical analyses for qPCR data with two treatment groups were determined by unpaired two-tailed *t*-test, while those with greater than two treatment groups were determined by one-way analysis of variance (ANOVA) with Holm–Sidak method for multiple comparisons. Statistical analyses for protein synthesis assays were determined by one-way ANOVA with Holm–Sidak method for multiple comparisons from values quantified with ImageJ software (US National Institutes of Health, Bethesda, MD). Statistical analyses for immunoblot quantification data with two treatment groups were determined by unpaired two-tailed *t*-test, while those with greater than two treatment groups were determined by one-way ANOVA with Holm–Sidak method for multiple comparisons. For proliferation assays, unpaired two-tailed *t*-test was used for comparisons to GFP-expressing cells. For cell death analysis, one-way ANOVA with Holm–Sidak method for multiple comparisons, summing the annexin V+/PI-, annexin V-/PI+, and annexin V+/PI+ populations for each conditions. For glutathione quantification of experiments with two conditions, an unpaired two-tailed *t*-test was performed. For remaining glutathione assays and all cystine and methionine uptake experiments, one-way ANOVA with Holm–Sidak method for multiple comparisons was used.

## Source data

The source data for the RNA-sequencing experiment can be found at GEO under the accession number GSE158605. The source data for *Figure 1C* can be found in *Figure 1—source data 1*. The source data for *Figure 2A* can be found in *Figure 2—source data 1*. The source data for NanoString heatmaps shown in *Figure 3* can be found in *Figure 3—source data 1*.

## Acknowledgements

We thank David J Kwiatkowski, Elizabeth P Henske, Randal J Kaufman, Alex Toker, and David M Sabatini for cell lines and materials and Min Yuan, Gerta Hoxhaj, and Issam Ben-Sahra for technical assistance and critical discussions. This research was supported by grants from the NIH: T32-ES016645 (MET), F31-CA228332 (MET), R35-CA197459 (BDM), and P01-CA120964 (BDM and JA); the Department of Defense's Congressionally Directed Medical Research Program on Tuberous Sclerosis Complex, award no. W81XWH-18-1-0659 (BDM); and a research grant from Zafgen (BDM and

JRM). BDM is a shareholder and scientific advisory board member of LAM Therapeutics and Navitor Pharmaceuticals. All other authors declare no competing financial interests.

## Additional information

### Competing interests
Brendan D Manning: Brendan Manning is a scientific advisory board member and stockholder of Navitor Pharmaceuticals and LAM Therapeutics. The other authors declare that no competing interests exist.

### Funding

| Funder | Grant reference number | Author |
|---|---|---|
| National Institutes of Health | R35-CA197459 | Brendan D Manning |
| National Institutes of Health | P01-CA120964 | John M Asara<br>Brendan D Manning |
| U.S. Department of Defense | W81XWH-18-1- 0659 | Brendan D Manning |
| National Institutes of Health | T32-ES016645 | Margaret E Torrence |
| National Institutes of Health | F31-CA228332 | Margaret E Torrence |
| Larimar Therapeutics | | Brendan D Manning<br>James R Mitchell |

The funders had no role in study design, data collection and interpretation, or the decision to submit the work for publication.

### Author contributions
Margaret E Torrence, Conceptualization, Data curation, Formal analysis, Validation, Investigation, Methodology, Writing - original draft; Michael R MacArthur, Data curation, Formal analysis, Visualization, Methodology; Aaron M Hosios, Conceptualization, Methodology; Alexander J Valvezan, Resources, Investigation; John M Asara, Data curation, Methodology; James R Mitchell, Supervision, Funding acquisition, Project administration; Brendan D Manning, Conceptualization, Formal analysis, Supervision, Funding acquisition, Project administration, Writing - review and editing

### Author ORCIDs
Margaret E Torrence (iD) https://orcid.org/0000-0003-2133-8599
Brendan D Manning (iD) https://orcid.org/0000-0003-3895-5956

### Ethics
Animal experimentation: All animal procedures were conducted under strict adherence to recommendations in the Guide for the Care and Use of Laboratory Animals of the National Institutes of Health and were approved by the Harvard Institutional Animal Care and Use Committee (#IS00000780).

### Decision letter and Author response
Decision letter https://doi.org/10.7554/eLife.63326.sa1
Author response https://doi.org/10.7554/eLife.63326.sa2

## Additional files

### Supplementary files
• Transparent reporting form

## Data availability

All data generated or analyzed during this study are included in the manuscript and supporting files. Source data files have been provided for Figures 1, 2, and 3. RNA-Seq data have been deposited in GEO under accession code GSE158605.

The following dataset was generated:

| Author(s) | Year | Dataset title | Dataset URL | Database and Identifier |
|---|---|---|---|---|
| MacArthur MR, Manning BD, Torrence ME | 2020 | The mTORC1-mediated activation of ATF4 promotes protein and glutathione synthesis (Tunicamycin) | https://www.ncbi.nlm.nih.gov/geo/query/acc.cgi?acc=GSE158605 | NCBI Gene Expression Omnibus, GSE158605 |

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
