## [Decision Letter]

**Acceptance summary:**

Torrence and colleagues explored the differential activation of ATF4 by the integrated stress response (ISR) and mTORC1. They presented strong genetic evidence that ISR and mTORC1 independently stimulate ATF4, and different sets of downstream targets are activated when ATF4 is stimulated by ISR or mTORC1. They further performed biochemical and metabolomic experiments to demonstrate that ATF4 triggered by mTORC1 induces protein synthesis and amino acid uptake, which is consistent with the canonical function of the mTORC1 pathway. This manuscript presents high quality data that support the major conclusion.

**Decision letter after peer review:**

Thank you for submitting your article "The mTORC1-mediated activation of ATF4 promotes protein and glutathione synthesis downstream of growth signals" for consideration by *eLife*. Your article has been reviewed by three peer reviewers, one of whom is a member of our Board of Reviewing Editors, and the evaluation has been overseen by David Ron as the Senior Editor.

The reviewers have discussed the reviews with one another and the Reviewing Editor has drafted this decision to help you prepare a revised submission.

Summary:

It is known that ATF4 plays a role in both the integrated stress response (ISR) and mTORC1 signaling. However, the relative roles of ATF4 in these two processes is not understood. Here the authors compare ISR and TORC1 responses of wild-type and ATF4^-/-^ cells. The authors conclude that mTORC1 stimulates only a sub-set of ATF4-dependent ISR target genes. These mTOR pathways include amino acid synthesis, uptake, and tRNA charging. The authors also report a role for mTORC1 and ATF4-dependent cystine uptake and glutathione synthesis. The data presented extend previous reports in the field (PMID: 26912861, PMID: 28494858). However, a number of points remain to be addressed to complete this study.

Essential revisions:

1) This study identifies an overlap between ATF4-dependent gene expression in response to the ISR and mTORC1 pathways. However, the analysis falls short of providing insight into the underlying mechanism. What accounts for the different subsets of ATF4-dependent gene expression under these conditions? The Discussion section of the manuscript proposes that it might be due to differences in ATF4 expression (high vs. low) or to the differential engagement of ATF4 heterodimeric partners, but no experimental tests of these predictions are presented. The difference could also be due to combinatorial actions of ATF4 with other transcription factors on gene promoters that are differentially activated by ISR vs. mTORC1. Comparative bioinformatics analysis of the promotors corresponding to the differentially expressed genes would be helpful. Moreover, biochemical analysis of ATF4 under these conditions is required, including analysis of heteromeric partners and DNA binding analysis.

2) Figure 2A and E. The authors need to explain the discordance between the RNA and protein levels. Densitometric analysis of the protein blots may be helpful. Moreover, the authors should show protein levels from the subset of "Amino acid transporter" genes as they did with "tRNA Charging" genes and "amino acid synthesis" genes in the Figure 2D and E.

3) MYC controls several amino acid transporters (PMID: 32022686). Since MYC can be activated downstream of mTORC1, the authors should examine MYC's contribution to amino acid transporter expression in the *Tsc2^-/-^* cells.

4) Figure 3D. The authors need to show the protein levels of the ATF4 targets (as they did in Figure 3C) to demonstrate resistance to rapamycin at the protein level. This is important because of the inconsistencies between RNA and protein expression shown in Figure 2A and E.

5) Figure 5B, C. The authors should show the protein levels of the Slc7a11 and *Slc3a2* to demonstrate the differential sensitivity to rapamycin in the different cell lines.

6) Figure 5E. The authors conclude that there is a critical role for mTORC1 in controlling cystine uptake through ATF4. However, the addition of cystine fully rescues cell growth in ATF4-depleted cells. How is that possible if the transporter levels would be reduced because of the lack of ATF4?

7) Figure 6H, the authors claim that glutathione levels were restored in *Atf4^-/-^* cells upon exogenous expression of ATF4 through the activation of *Slc7a11*. However, ATF4 also regulates the subunits of glutamate-cysteine ligase levels (PMID: 17297441). How are the Gclc and Gclm levels in *Atf4^-/-^* cells in Figure 6G and H? The authors should measure these proteins in total ATF4-KO instead of using siRNA (Figure 7—figure supplement 1E).

---

## [Author Response]

Essential revisions:1) This study identifies an overlap between ATF4-dependent gene expression in response to the ISR and mTORC1 pathways. However, the analysis falls short of providing insight into the underlying mechanism. What accounts for the different subsets of ATF4-dependent gene expression under these conditions? The Discussion section of the manuscript proposes that it might be due to differences in ATF4 expression (high vs. low) or to the differential engagement of ATF4 heterodimeric partners, but no experimental tests of these predictions are presented. The difference could also be due to combinatorial actions of ATF4 with other transcription factors on gene promoters that are differentially activated by ISR vs. mTORC1. Comparative bioinformatics analysis of the promotors corresponding to the differentially expressed genes would be helpful. Moreover, biochemical analysis of ATF4 under these conditions is required, including analysis of heteromeric partners and DNA binding analysis.

This important question is unknown for the majority of studies on ATF4, which can be activated by many distinct upstream stress signals, in addition to growth factor signaling through mTORC1, to elicit both specific and overlapping gene responses. Given that the ATF4-dependent targets found to be shared between the ISR and mTORC1 signaling in our analyses represent many of the canonically reported ATF4 targets in other settings, we hypothesized and discussed in our original submission that these might represent those targets that are most sensitive to induction of ATF4 protein levels. Consistent with this idea, an analysis of the 774 genes significantly induced by tunicamycin in an ATF4-dependent manner (Figure 1—source data 1) reveal that the 61 genes also significantly induced by insulin and sensitive to rapamycin are greatly enriched amongst the top-100 most significantly induced by tunicamycin (depicted in red in the new Figure 1E, with individual genes denoted in Figure 1—source data 1).

New Figure 2: As suggested, we also used two bioinformatic approaches to analyze the promoters of these ATF4-dependent gene targets. First, we used the CiiiDER tool to predict DNA-binding motifs of transcription factors in the promoters of the 61 ATF4-dependent genes shared between the ISR and mTORC1 signaling versus the top-200 ISR only genes. This worked quite well and revealed that C/EBP-binding motifs are the most enriched amongst the shared genes, while binding motifs for the TEAD family of transcription factors were enriched for the ISR only genes (new Figure 2A). This latter observation is interesting in light of recent studies indicating that TEAD and its binding partners YAP/TAZ, downstream targets of the Hippo pathway, are activated by ER stress and can engage UPR targets (PMID: 25695629; PMID: 31558567). Our second unbiased analysis employed the Cistrome database of published and curated genome-wide ChIP-seq studies to determine whether specific transcription factors have been found to engage promoters of the 61 genes with shared regulation. Once again, the C/EBP family of transcription factors were found among ChIP-seq studies to most commonly bind to the promoters of these genes (new Figure 2B).

ATF4 can heterodimerize with the entire C/EBP family (PMID: 12805554). Thus, to follow up on our bioinformatic observations with functional assays, we used siRNA-mediated knockdown of individual C/EBP isoforms (α, β, δ, γ) and ATF4 and examined expression of representative genes from distinct functional categories (*Mthfd2*, *Slc7a5*, *Aars*) from the list of 61 ATF4-dependent targets shared in their regulation by mTORC1 and the ISR. This analysis revealed a high degree of cross regulation between these different isoforms and ATF4 in regulation of their own transcripts, which made it difficult to assess the direct effects of one or more of these C/EBP family members on downstream targets (new Figure 2C). Unfortunately, we were unable to identify reliable isoform-specific C/EBP antibodies that gave clear and reproducible signals by immunoblot or IP from MEF lysates, with antibodies tested using siRNA-mediated knockdowns as controls for the proper bands. To varying degrees, C/EBPβ, δ, and γ knockdowns all decreased expression of the targets tested, but only C/EBPγ did so for every gene without influencing ATF4 protein levels (new Figure 2C, D). Interestingly, many of the 61 shared genes identified overlap with those found previously to be regulated by ATF4 through heterodimerization with C/EBPγ in response to amino acid deprivation stress (PMID: 26667036). Thus, we used siRNA-mediated knockdown of ATF4 or C/EBPγ to determine the effects on the insulin and rapamycin response of these representative targets (new Figure 2D). These data demonstrated that C/EBPγ was required for induction of these genes by insulin and was also required for the insulin-stimulated increase in ATF4 gene expression. Together, our new bioinformatic analyses and functional data suggest that C/EBPγ engagement is responsible for at least part of the ATF4-dependent gene expression program shared between mTORC1 signaling and the ISR. However, as shown by use of the rapamycin-resistant ATF4 cDNA, the mTORC1-mediated regulation of these gene targets is primarily through effects on ATF4 protein levels.

2) Figure 2A and E. The authors need to explain the discordance between the RNA and protein levels. Densitometric analysis of the protein blots may be helpful. Moreover, the authors should show protein levels from the subset of "Amino acid transporter" genes as they did with "tRNA Charging" genes and "amino acid synthesis" genes in the Figure 2D and E.

Transcript and protein levels are often in discordance, given that protein abundance is independently influenced by varying rates of translation, stability, and degradation. Quantification of these blots was facilitated by use of a LICOR Odyssey Imaging System, which provides linear quantification rather than that derived from enzyme-linked blotting methods. Corresponding protein quantification of the blots in Figure 2D-E is now provided as new Figure 3—figure supplement 1C-D. We tested antibodies, together with siRNA controls for specificity, for multiple amino acid transporters found to be regulated at the transcript level by mTORC1 signaling through ATF4, including SLC7A5, SLC3A2/CD98, SLC1A5, and SLC7A11. Some of these antibodies work for blotting of human protein extracts, but unfortunately, only SLC7A11 gave a specific band that was decreased with siRNAs in MEFs (new Figure 6—figure supplement 1A), see comment 5 below.

3) MYC controls several amino acid transporters (PMID: 32022686). Since MYC can be activated downstream of mTORC1, the authors should examine MYC's contribution to amino acid transporter expression in the Tsc2^-/-^ cells.

We expanded our analysis of ATF4 gene targets to include both amino acid transporters (*Slc7a11*, *Slc7a5*, *Slc3a2*, and *Slc1a5*) and genes of non-essential amino acid synthesis (*Psat1* and *Mthfd2*), some of which have been found to be regulated by Myc in other settings. While a few of these were found to be significantly reduced with siRNA-mediated knockdown of c-Myc, in all cases ATF4 knockdown more strongly and significantly reduced their expression (new Figure 3—figure supplement 1L).

4) Figure 3D. The authors need to show the protein levels of the ATF4 targets (as they did in Figure 3C) to demonstrate resistance to rapamycin at the protein level. This is important because of the inconsistencies between RNA and protein expression shown in Figure 2A and E.

As mentioned with comment 2 above, we do not expect the RNA and protein expression to match in degree of changes. However, we have confirmed by immunoblot that the levels of proteins encoded by specific ATF4 gene targets are indeed elevated and rapamycin resistant in ATF4 knockout cells reconstituted with the ATF4 cDNA lacking its 5’UTR, but not the ATF4-DBD mutant version of this cDNA (new Figure 4—figure supplement 1).

5) Figure 5B, C. The authors should show the protein levels of the Slc7a11 and Slc3a2 to demonstrate the differential sensitivity to rapamycin in the different cell lines.

After testing several different commercially available antibodies, we were able to find SLC7a11 antibodies that specifically and reliably recognized this protein in either mouse or human cell protein extracts, and an SLC3A2/CD98 antibody that works for human proteins. SiRNA validation of the SLC7A11 antibody in MEFs is provided in new Figure 6—figure supplement 1A. Consistent with the mTORC1-regulated changes in transcript levels described in mouse and human lines in the original manuscript, we find that mTOR inhibitors significantly decrease SLC7A11 protein levels in *Tsc2^-/-^* MEFs (new Figure 6D and quantified in new Figure 6—figure supplement 1B), and in wild-type MEFs, insulin stimulates an increase in SLC7A11 protein in a manner that is sensitive to mTOR inhibitors (new Figure 6E and quantified in new Figure 6—figure supplement 1C). The protein levels of SLC7A11, but not SLC3A2/CD98, are also significantly sensitive to mTOR inhibitors in human LNCaP and PC3 cells (new Figure 6F and Figure 6—figure supplement 1F-H). As above, these blots were quantified using the LICOR system.

6) Figure 5E. The authors conclude that there is a critical role for mTORC1 in controlling cystine uptake through ATF4. However, the addition of cystine fully rescues cell growth in ATF4-depleted cells. How is that possible if the transporter levels would be reduced because of the lack of ATF4?

There is apparent confusion between cysteine (Cys) and cystine (Cys_2_) reflected by this comment. The figure (now Figure 6H) and accompanying text describe rescue of proliferation by cysteine but not cystine, as also observed in Figure 7F for glutathione levels. As noted in the text, cysteine gets into cells through neutral amino acid transporters that are not regulated by ATF4, as the SLC7A11-dependent transport of cystine is.

7) Figure 6H, the authors claim that glutathione levels were restored in Atf4^-/-^ cells upon exogenous expression of ATF4 through the activation of Slc7a11. However, ATF4 also regulates the subunits of glutamate-cysteine ligase levels (PMID: 17297441). How are the Gclc and Gclm levels in Atf4^-/-^ cells in Figure 6G and H? The authors should measure these proteins in total ATF4-KO instead of using siRNA (Figure 7—figure supplement 1E).

Consistent with our data shown regarding rapamycin-sensitive expression of Gclc and Gclm transcripts, which were unaffected by siRNA-mediated knockdown of ATF4 (Figure 7—figure supplement 1D-E), only minor changes in GCLC and GCLM protein levels were detected with ATF4 knockout in wild-type cells or knockout and reconstitution in *Tsc2^-/-^* cells (Figure 7—figure supplement 1F-G). Insulin stimulated a modest increase in the levels of these proteins in both wild-type and ATF4 knockout MEFs, in a manner sensitive to mTOR inhibitors, but mTORC1-mediated effects on these proteins were not apparent in *Tsc2^-/-^* cells with or without ATF4. Thus, in these settings, we do not observe reproducible effects on levels of GCL enzyme complex components that would explain the observed effects of mTORC1 signaling, ATF4, SLC7A11, and cysteine on intracellular glutathione abundance, detailed in Figure 7.